# Numerical quantitation on the effect of coating materials on the mixing state retrieval accuracy of fractal black carbon based on single particle soot photometer

Jia Liu[1, 2, 3], Donghui Zhou[2, 3], Guangya Wang[2, 3], Cancan Zhu[2, 3], and Xuehai Zhang[4]

[1]Department of Atmospheric and Oceanic Sciences and Institutes of Atmospheric Sciences, Fudan University, Shanghai 200438, China.
[2]Engineering Research Center of Zero-carbon Energy Buildings and Measurement Techniques, Ministry of Education, Hebei University, Baoding 071002, China.
[3]China Meteorological Administration Xiong'an Atmospheric Boundary Layer Key Laboratory, Baoding 071800, China.
[4]School of Information Science and Engineering, Henan University of Technology, Zhengzhou 450001, China.

*Correspondence*: Jia Liu (liujia@hbu.edu.cn)

**Abstract.** The mixing states of coated black carbon particles (BC) are essential for assessing their optical properties and radiative effects. The single particle soot photometer (SP2) measures BC mixing states based on the assumption that the refractive index (RI) of the coating material is constant $1.50+0i$. However, atmospheric BC aerosols may be coated by diverse materials with various refractive indices, therefore, the unrealistic assumption of coating RI would bring errors in the measured mixing states. For the evaluation of the effects of coating materials on the measurement accuracy of BC mixing states, from the perspective of numerical simulation, typical materials: sulfate, non-absorbing organic carbon (OC), and brown carbon (BrC) are selected as coatings of BC in this study. The fractal closed-cell model and coated-aggregate model are constructed to represent BC with thin and thick coatings, multiple sphere T-matrix (MSTM) simulation results are used to study the optical properties of realistic coated BC, and the mixing states are optically retrieved based on Mie theory according to the retrieval principle of the SP2. Results showed that the SP2 would underestimate mixing states for fractal BC with BC core diameters larger than about 200 nm, and the relative errors of retrieved mixing states for BC coated by sulfate and OC are evidently larger than those for BC coated by BrC. Furthermore, when the SP2 measured mixing states are employed to estimate the absorption enhancement ($E_{ab}$) and radiative forcing of BC aerosols, the estimation error of $E_{ab}$ for BC with BrC coating is significantly larger than that with sulfate and OC coatings, and the estimation $E_{ab}$ could be up to 6.19 times higher than the inherent value of fractal BC. Compared with fractal BC, the radiative forcing estimation error at 1064 nm for BrC-coated BC can reach -88.73%, while the estimation errors of BC coated by the other two materials range from -65.22% to -37.50%. This study highlights that the coating materials and fractal morphology should be considered when the SP2 is used to measure the mixing state of BC.

## 1. Introduction

Black carbon (BC) is an important component of atmospheric aerosol, which is mainly generated from the incomplete combustion of carbon-rich materials such as fossil and biomass (Bond et al., 2013). Both natural sources, like volcanic eruptions, and anthropogenic sources, like heating and cooking, emit a considerable amount of BC aerosols into the atmosphere (Murphy et al., 2014; Santoso et al., 2020). Although BC accounts for a small proportion of atmospheric aerosol, as an optical absorptive aerosol, its optical absorption properties from the visible to the infrared wavelengths lead to a strong positive forcing on climate change (Li et al., 2019). BC also prevents the vertical diffusion of pollutants by altering the atmospheric temperature structure and accelerates the melting of snow and ice when it deposits on their surfaces (Ding et al., 2016). Optically absorption-dominated fresh BC particles are usually present as loose chain-like aggregates consisting of numerous near-spherical monomers (Ceolato et al., 2020). During the aging in the atmosphere, fresh BC particles can be coated by various chemical components such as sulfate, non-absorbing organic carbon (OC), and brown carbon (BrC) through condensation and coagulation, forming aged BC aerosols with complex mixing structures (China et al., 2013). The heterogeneous mixing has a complicated and profound influence on the radiative effect of BC aerosols.

The "mixing state" is a key microphysical property for aged BC, describing the mixing structure of BC and its coating. It can be characterized through different principles and instruments, such as the single particle soot photometer (SP2), single particle aerosol mass spectrometry (SPAMS), and single-particle soot mass spectrometry (SP-AMS) (Liu et al., 2023b; Liu et al., 2025). The SP2 (Droplet Measurement Technology, Inc.), as one of the most effective online instruments that measures the mixing state of coated BC based on the combination of laser-induced incandescence and light scattering measurement, has been widely employed in laboratory and field observations (Liu et al., 2020a; Liu et al., 2022; Schwarz, 2019). The optical cavity of the SP2 consists of a laser with a wavelength of 1064 nm and two pairs of symmetrically arranged detectors relative to the laser, and the detectors cover viewing angles of 13-77° and 103-167°, respectively (Gao et al., 2007). When the coated BC passes through the laser beam vertically, the scattered signal is constantly recorded by the scattering detectors. As the coatings are heated and evaporated, the original Gaussian pattern of the scattered signal is dynamically disturbed. Through the extraction of the leading edge of the original scattered signal when the coated BC initially enters the light beam, the complete Gaussian scattering signal can be reconstructed based on the Leading Edge Only (LEO) method (Gao et al. 2007). The differential scattering property can be derived, and the optical equivalent diameter of the whole coated BC particle ($D_p$) can be retrieved based on the Mie scattering theory for the spherical core-shell model. The refractive indices of BC and coating used by Mie theory are roughly set to constant 2.26+1.16$i$ and 1.50+0$i$, respectively (Schwarz et al., 2006). After the coating is completely evaporated, the bare BC emits incandescent signals. Since the peak of the incandescent signal is proportional to BC mass, the volume equivalent diameter of BC core ($D_c$) can be obtained by assuming spherical particles and using BC density of 1.80 g/cm$^3$ (Schwarz et al., 2006; Bond and Bergstrom, 2006). Finally, the mixing state of coated BC at the single particle level can be characterized by the SP2 as the particle diameter ratio of the whole particles to the BC core ($D_p/D_c$)(Schwarz et al., 2008a;

Schwarz et al., 2008b). In addition, through the statistical analyses of the lag time between the appearance of the scattered signal and the incandescent signal, the particles can be classified into bare-to-thinly coated BC and heavily coated BC. However, since the fractal morphologies, coating structures, and coating materials of aged BC aerosol are complicated and diverse, the spherical core-shell morphological model and a constant refractive index $1.50+0i$ of coatings employed during the retrieval of mixing state are oversimplified compared with the realistic situations. More specifically, the BC core and outer coating are assumed to be concentric double spheres, while the realistic BC can be an aggregate covered by a film (thinly-coated) or encapsulated in a shell (heavily-coated) (Qin et al., 2022). The morphology simplification of aged BC would result in inherent errors in the $D_p$ and the mixing state. Our previous studies have preliminarily revealed that fractal morphology and coating structure can result in mixing state retrieval errors up to approximately 80%, and it is worth noting that the characterization of $D_p/D_c$ based on Mie theory would miss some amount of mixing state results for coated BC with certain microphysical parameters (Liu et al., 2023b). Besides, the coatings of atmospheric BC aerosols can be inorganic salts like sulfate or organics with optical absorptive or non-absorptive properties. Both the real and imaginary parts of the refractive indices of these components are various and further affect the optical properties of coated BC (Zhang et al., 2021). Nevertheless, the refractive index of the coating shell is roughly assumed to be constant $1.50+0i$ in the current optical retrieval scheme of the SP2, so the single refractive index would also lead to retrieval errors in both the $D_p$ and $D_p/D_c$. Given this, what are the quantitative effects of the oversimplification of both morphology and coating refractive index on the retrieval errors in mixing states? Furthermore, as a widely used instrument characterizing mixing state, the measurements of the SP2 are usually further used to evaluate the critical parameters, like absorption enhancement and radiative effect of aged BC. Then, how do the inherent errors in mixing states affect the evaluation accuracy of optical and radiative parameters?

To answer the above questions, numerical simulations are conducted in this study. State-of-the-art closed-cell model (CCM) and coated-aggregate model (CAM) are built to represent typical thinly and heavily coated BC, and these two kinds of BC can be classified using the lag time measured by the SP2. Three materials, that is, sulfate, non-absorbing organic carbon (OC), and brown carbon (BrC), are employed as typical coating components of BC aerosols. The optical properties of fractal BC models are simulated by multiple sphere T-matrix (MSTM) methods and regarded as the optical properties of realistic coated BC. The optical equivalent diameter and mixing state of coated BC are retrieved using differential scattering properties, based on Mie theory and the core-shell model, according to the operating principle of the SP2. Then, the optically retrieved results of $D_p/D_c$ are compared with the preset values of fractal models, and the effects of coating materials and coating structures on the retrieval accuracies of BC mixing states are quantified. Furthermore, to promote the reasonable employment of measured results of the SP2, the prediction performances of absorption enhancement ($E_{ab}$) and radiative forcing ($RF$) of BC are evaluated based on the retrieved $D_p/D_c$.

## 2. Methodology

**2.1 Morphological models of BC aerosols**

The freshly emitted bare BC particles are fractal clusters that consist of a large number of near-spherical monomers, following the scaling rule (China et al., 2013; Liu et al., 2020b; Wang et al., 2021b):

$$N_s = k_f \left( \frac{R_g}{a_0} \right)^{D_f} , \tag{1}$$

$$R_g = \sqrt{\frac{1}{N_s} \sum_{i=1}^{N_s} r_i^2} , \tag{2}$$

where $a_0$ is the monomer radius, $N_s$ is the monomer number, $k_f$ and $D_f$ are pre-factor and fractal dimension that control the morphology of BC aggregate, $R_g$ is the gyration radius that describes the spatial scale of BC, and $r_i$ is the distance between the mass center of the $i$-th monomer and the mass center of the whole aggregate (Zeng et al., 2019; Kahnert and Kanngiesser, 2020). In general, the newly emitted BC particles show loose chain structures, with the value of $D_f$ being about 1.80. During the atmospheric aging process, the fractal structure becomes more compact and spherical with $D_f$ increasing to about 2.80 and even larger. In this study, the pre-factor is assumed to be 1.20, and the monomer radius of BC is fixed to 20 nm, which are typical values observed in experiments (Sorensen and Roberts, 1997). The monomer number is set to vary in the range of 50-2000 with a step of 50, thus covering almost all the size distribution of atmospheric BC. Then, the volume equivalent diameter of BC ($D_c$) can be obtained:

$$D_c = 2a_0 \sqrt[3]{N_s} \tag{3}$$

Furthermore, bare BC tends to be coated by different atmospheric components during the aging process and forms mixed particles with complicated coating structures, such as thinly film-coated BC and heavily encapsulated BC. Typical coating components are light-absorptive brown carbon (BrC) and non-absorptive materials like organic matter and inorganic salt (Kholghy et al., 2013). Two fractal morphological models, closed-cell model (CCM) with $D_f$=1.80, 2.20, and 2.60, and coated-aggregate model (CAM) with $D_f$=2.60 and 2.80, are built to represent the non-spherical mixing structures of BC with thin and heavy coatings, respectively, as shown in Figure 1. The amount of coating material in aged BC aerosol is expressed by the volume fraction of BC core ($V_f$):

$$V_f = \frac{V_{BC}}{V_{total}} \tag{4}$$

where $V_{BC}$ and $V_{total}$ are the volumes of BC core and coated BC, respectively. Furthermore, in the measurement framework of SP2, the amount of coating in the whole particle can also be described by the mixing state, which is the ratio of the equivalent sphere diameters of coated BC ($D_p$) and BC core ($D_c$):

$$\frac{D_p}{D_c} = \frac{1}{\sqrt[3]{V_f}}$$

(5)

For constructing complicated morphological models of coated BC, the Diffusion-Limited Aggregation (DLA) algorithm developed by Wozniak et al. (2012) was first employed to generate bare BC fractal aggregates. The CCM is also a fractal aggregate whose monomer is a concentric double-layer sphere, the inner sphere represents BC, and the outer sphere represents the coating. For the construction of the CCM, the original fractal aggregate generated by DLA is enlarged according to the BC volume fraction, and then a soot sphere is added into each enlarged monomer. As for the coated aggregate model (CAM), the whole fractal aggregate generated by DLA is encapsulated using a spherical coating (Liu et al., 2023a). The overall morphological parameters for CCM and CAM are summarized in Table 1.

## 2.2 Numerical simulation of optical and radiative properties

In this study, optical properties such as optical cross-section and scattering matrix at 1064 nm, which is consistent with the observation wavelength of the SP2, are simulated using the multiple sphere T-matrix (MSTM) method developed by Mackowski and Mishchenko (2011). Accordingly, the real and imaginary parts of the complex refractive index of the BC core are set to 2.26 and 1.26, respectively. The density of the BC core ($\rho_{BC}$) is set to 1.80 g/cm$^3$ (Liu et al., 2014; Zhang et al., 2019). Sulfate, non-absorbing organic carbon (OC), and absorbing brown carbon (BrC), are selected as representative coatings around the BC core. The complex refractive indices are set as $1.51+10^{-5}i$, $1.52+0.02i$, and $1.55+0.18i$, and 1.77 g/cm$^3$, 1.20 g/cm$^3$, and 1.20 g/cm$^3$ are selected as typical density values for these three components (Wang et al., 2021a; Luo et al., 2021).

For the quantification of the effects of coating materials on the optical retrieval errors of the SP2, the differential scattering cross-sections of the CAM and CCM with preset mixing state ($D_{p,v}/D_{c,v}$) are recognized as representative observations for realistic coated BC particles. According to the optical retrieval principle of the SP2, a series of core-shell models with the refractive index and density of the shell are set to $1.50+0i$ and 1.20 g/cm$^3$, the diameter of the core is the same as the volume equivalent sphere of the BC core. The differential scattering cross-sections of core-shell models are simulated using the Mie scattering theory. The optical retrieval of the mixing state ($D_p/D_c$) of coated BC is essentially searching for the core-shell model whose optical property is closest to the optical property of prebuilt CAM or CCM. With the variation of particle diameter $D_p$, when the distinctions between the differential scattering cross-sections of fractal models and core-shell models are the minimum based on the ordinary least squares method, the corresponding $D_p$ is the retrieval result, and the mixing state ($D_p/D_c$) can be further obtained. The relative difference $\chi^2$ between the differential scattering cross-sections for fractal BC and core-shell models, which is used in the retrieval, is defined as follows:

$$\chi^2 = \left[ \frac{dC_{sca-\text{Mie}}\left(D_p\right) - dC_{sca-\text{MSTM}}\left(D_{p,v}\right)}{d\,C_{sca-\text{MSTM}}\left(D_{p,v}\right)} \right]^2 \tag{6}$$

where $dC_{sca\text{-MSTM}}$ and $dC_{sca\text{-Mie}}$ are the differential scattering cross-sections of the fractal BC model and core-shell model, respectively. In addition, the relative error (*RE*) of the retrieved mixing state is calculated:

$$RE = \frac{D_p/D_c - D_{p,v}/D_{c,v}}{D_{p,v}/D_{c,v}} \times 100\% \tag{7}$$

The mixing state ($D_p/D_c$) is an important parameter for the aged BC aerosols and can be further used to evaluate the optical absorption and radiative effect of BC caused by the coatings. The improper assumptions of both the aerosol morphology and the refractive index of the coating would result in retrieval errors in $D_p/D_c$ using the SP2. To facilitate the application of the SP2 measurement results in optical and radiative simulation research, the precision in the estimated typical optical and radiative properties caused by the retrieval errors in $D_p/D_c$ should also be quantified numerically. The absorption enhancement ($E_{ab}$) is usually used to depict the increase in absorption cross-section due to the growth of the coating amount:

$$E_{ab} = \frac{C_{abs,p}}{C_{abs,c}} \tag{8}$$

where the $C_{abs,p}$ and $C_{abs,c}$ are the absorption cross-sections for the whole coated BC particle and the BC core, respectively. As for the radiative property, the simple forcing efficiency (SFE), which is defined as the radiative forcing normalized by BC mass and represents the amount of solar energy added to the Earth's atmospheric system by the given BC mass, is used to evaluate the radiative effect of coated BC aerosol Bond and Bergstrom (2006). The differential form of simple forcing efficiency $dSFE/d\lambda$ could be simulated according to the following equation:

$$\frac{dSFE}{d\lambda} = -\frac{1}{4}\frac{dS(\lambda)}{d\lambda}\tau^2\left(1-F_c\right)\left[2\left(1-a_s\right)^2\beta(\lambda)\times MSC(\lambda) - 4a_s MAC(\lambda)\right] \tag{9}$$

where $\lambda$ is the wavelength, the *MSC* and *MAC* are the mass scattering cross-section and mass absorption cross-section, respectively. The $dS(\lambda)/d\lambda$ is the spectral solar irradiance from ASTM G173-03 (ASTM G173-03 Reference Spectra https://www.nrel.gov/grid/solar-resource/spectra.html), the $\tau$ is atmospheric transmittance set to 0.79 (Bhandari et al., 2019), the $F_c$ is cloud fraction set to 0.60 (Bond and Bergstrom, 2006), the $a_s$ is surface albedo set to 0.19 for typical urban surface (Liu et al., 2023b), and the $\beta$ is backscattering fraction set to 0.15 (Peng et al., 2022). The *MSC* and *MAC* could be calculated with the help of particle mass and coating volume:

$$MAC = \frac{C_{abs}}{m} \tag{10}$$

$$MSC = \frac{C_{sca}}{m} \tag{11}$$

$$m = \rho_{BC}V_{BC} + \rho_{coating}V_{coating} \tag{12}$$

$$V_{coating} = \left(1-V_f\right)V_{BC}\Big/V_f \tag{13}$$

where $m$ is the mass of the whole coated BC particle, $C_{abs}$ and $C_{sca}$ are the absorption cross-section and scattering cross-section for the coated BC, respectively. $\rho_{BC}$ and $\rho_{coating}$ are the density of the BC core and the coating, respectively. $V_{coating}$ is the volume of the coating.

## 3. Result and discussion

### 3.1 Influence of different coating materials on the retrieval of mixing states

When the BC is coated by different atmospheric components such as sulfate, OC, and BrC, the optical properties of coated particles change accordingly. The variation of complex refractive indices of coatings is one of the fundamental reasons for the optical differences. As for coating materials, the real part of the complex refractive index refers to the ratio of the propagation speed of light in a vacuum to that in the coatings, reflecting the scattering ability of the coating material. The imaginary part of the complex refractive index refers to the attenuation of light during propagation, which reflects the absorption ability of

the coating material (Mishchenko et al., 2002). Therefore, even for coated BC with the same morphological parameters, the optical differences caused by coating materials will inevitably lead to variations in the optical retrieved mixing states using the SP2, which should be quantified.

**Figure 1** shows the retrieved results of $D_p/D_c$ for BC particles coated by sulfate in different morphological models. The dashed lines stand for the preset values of $D_{p,v}/D_{c,v}$ when constructing fractal BC models. As can be seen from Fig.1 (a) and (b) that

the retrieved $D_p/D_c$ for thinly coated BC decreases with the diameter of the BC core. The retrieved mixing states for coated BC with core diameters larger than 200 nm are smaller than the preset values, indicating that the SP2 will underestimate the mixing states for coated BC whose core is larger than 200 nm. On the contrary, when the BC core is smaller than 200 nm, the SP2 will overestimate the mixing states for thinly coated BC. With the increase of the fractal dimension, the fractal closed-cell structure becomes more compact, and the retrieval errors in $D_p/D_c$ decrease. The increase of BC volume fraction due to a small

amount of coatings also leads to reduced values of $RE$. Fig. 1 (c) and (d) indicate that the values of $RE$ for thinly sulfate-coated BC particles with $D_f$=2.60 represented by the CCM are larger than those of thickly coated BC particles represented by the CAM, especially for larger BC volume fraction, which reveals that the SP2 has a better performance in characterizing the mixing state for BC thickly coated by sulfate. The $RE$ in the retrieved $D_p/D_c$ for the thickly coated BC becomes more remarkable with the increase of $D_f$. It should be noted that there are a certain number of missing data points of retrieved $D_p/D_c$,

because of the inherent differences between the fractal BC model and the spherical core-shell model. More specifically, there is no core-shell model whose differential optical properties could match the coated BC particle with certain parameters corresponding to the missed points. There are missed values of retrieved $D_p/D_c$ for sulfate-coated both CCM and CAM with BC core diameter ranges from about 200 to 400 nm, as shown by Liu et al. (2023b). With the fractal dimension increasing from 2.60 to 2.80, the BC core becomes a sphere, and the coated-BC is closer to the core-shell model, optical distinctions between the fractal BC and the core-shell model are smaller, which reduces the missed points of retrieved $D_p/D_c$. This means that when the SP2 is employed to characterize atmospheric BC aerosols, even if it can effectively monitor the differential scattering properties of sulfate coated BC, it may still lose the observation results of the mixing states for particles whose BC core is distributed in a certain size range due to the oversimplification of the morphological model, and it may lead to a bias toward size ranges of coated BC that the SP2 can characterize when considering BC particle ensemble.

**Figure 2** shows the retrieved results of the mixing state for thinly coated BC particles with different coating materials using the closed-cell model. It is obvious that when the coated BC aerosols show relatively loose fractal structures, the retrieved mixing state is not very sensitive to the coating components, thus the retrieved results of $D_p/D_c$ for BC particles coated by sulfate, OC, and BrC are all similar, even though $D_p/D_c$ for BrC-coated BC is slightly smaller than that for the other two components. Essentially, the BC aerosols with different coating materials have almost identical optical properties, including differential scattering properties observed by the SP2, due to the joint influence of refractive index and coating structure. On the one hand, the distinctions in both the real and imaginary parts of the refractive indices of all these coatings at 1064 nm are not very significant, compared with the refractive index of the BC. On the other hand, when the soot cores are distributed in each of the coating monomers in the closed-cell model, the interaction of each soot would be weakened due to the isolation of the coatings. Furthermore, the lensing effect, which means the enlargement of optical properties caused by the coating, would also be unobvious when the volume fraction of BC decreases to a certain degree in this study, like 0.075 and 0.05 (Lack and Cappa, 2010). This is also the reason why the retrieved results of $D_p/D_c$ for coated BC are comparable with the $V_f$ increases from 0.05 to 0.075. When the fractal dimension $D_f$ increases, the retrieved $D_p/D_c$ also increases obviously, and the corresponding retrieval error is slightly reduced, especially for BC cores smaller than about 400 nm, which account for a significant portion of atmospheric BC. However, compared with the preset $D_{p,v}/D_{c,v}$, the optically retrieved $D_p/D_c$ for most coated BC are underestimated, and the largest retrieval error could be about -63.16%. In short, the SP2 is not sensitive to the coating components, such as sulfate, OC, and BrC, for thinly coated BC when it is used to characterize the mixing state.

**Figure 3** shows the retrieved results of the mixing state for thickly coated BC particles with different coating materials using the coated-aggregate model. With the diameter increase of the BC core, the retrieved $D_p/D_c$ gradually converges to 1. Different from the thinly coated BC particles, the retrieval results of the mixing states for the BC particles coated by sulfate and OC with thick coatings no longer coincide very closely but show slight differences. However, the retrieval results of the mixing state of BC particles heavily coated by BrC are significantly different from those of BC particles coated by sulfate or OC, and the retrieved $D_p/D_c$ are higher compared with the other two coatings. Since the coated-aggregate model consists of a spherical coating and a soot cluster, the interaction between each soot monomer is more intense than in the closed-cell model, and the

lensing effect of the coating is more obvious. Therefore, the influence of the refractive index on the optical properties of coated

BC is remarkable. The minor differences in the refractive indices between sulfate and OC could explain the similarity in the retrieved $D_p/D_c$, while the relatively large differences in the refractive indices between BrC and the other two coatings may have caused much more significant distinctions in retrieved $D_p/D_c$. With the increase of the diameter of the BC core, the retrieved mixing states are overestimated at first and then underestimated, and the diameters of the turning points range from 200 nm to 300 nm. The mixing states for a large part of the coated BC particles are underestimated, but the turning points

correspond to larger diameters with the decrease of the preset volume fraction of BC core. In addition, the differences between the retrieval results of the mixing state of BC particles heavily coated by BrC and that of BC particles heavily coated by sulfate and OC increase with the increase of both fractal dimension and preset BC volume fraction. The maximum retrieval error of the mixing state for BC particles thickly coated by three different materials could reach -63.04% when BC is coated by sulfate with a fractal dimension of 2.60. Overall, the retrieval errors in $D_p/D_c$ for BrC-coated BC are smaller, which illustrates that the

SP2 measures the mixing states of BC particles thickly coated by BrC with a little higher accuracy than that of BC coated by other materials. Nevertheless, there are more missed points of retrieved $D_p/D_c$ for the optically absorbing coating components, indicating that the coupling effect of both morphological simplification and single refractive index of coating in the optical retrieval of the SP2 would lead to the loss of more observation results of the mixing states, particularly for thickly coated BC aerosol. Therefore, both the complex non-spherical morphology of coated BC and the diversified coating components should

be considered to improve the retrieval accuracy of BC mixing state based on the SP2.

For further understanding of the distribution of the retrieved mixing state of all the coated BC with the fractal dimension of 2.60, a half-violin diagram is drawn and shown in **Figure 4**. Three distinct regions, colored in purple, yellow, and red, represent the distribution of the retrieved $D_p/D_c$ for black carbon particles coated by sulfate, OC, and BrC, respectively. It can be seen from the figure that the distributions of the retrieved $D_p/D_c$ for BC particles coated by sulfate and OC are almost the same. The

averaged retrieval results of $D_p/D_c$ increase with the increase of the preset volume fraction of BC, but there are still large discrepancies between the preset $D_{p,v}/D_{c,v}$ and the retrieved $D_p/D_c$. For coated BC with the same $D_{p,v}/D_{c,v}$, the average value of retrieved $D_p/D_c$ for thinly coated BC particles represented by the CCM is smaller than that of the thickly coated BC particles represented by the CAM. However, the distribution width of the retrieved $D_p/D_c$ for BC particles coated by BrC is markedly narrower than that of BC particles coated by the other two materials. This is because there are more missed points of the

retrieved $D_p/D_c$ for BrC-coated BC. In addition, the averaged retrieval results of $D_p/D_c$ for BC particles thinly coated by BrC increase with the increase of the preset $D_{p,v}/D_{c,v}$, while the averaged $D_p/D_c$ for thickly coated BC particles decrease with the increase of the preset $D_{p,v}/D_{c,v}$, which makes the retrieval error much larger.

Atmospheric BC aerosols mostly exist as particle groups that follow certain size distributions. The retrieval results (RR) of the mixing states for BC particle groups coated by different components and the corresponding retrieval errors (RE) are analyzed,

under the assumption that the volume equivalent diameter of BC follows the typical lognormal distribution:

$$n(d) = \frac{1}{\sqrt{2\pi}d\ln(\sigma_g)}\exp\left[-\left(\frac{\ln(d)-\ln(d_g)}{\sqrt{2}\ln(\sigma_g)}\right)^2\right] \tag{14}$$

where $d$ is the diameter of BC, $d_g$ and $\sigma_g$ are the geometric mean diameter and standard deviation, which are set to 0.15 μm and 1.59 in this study, respectively (Yu and Luo, 2009; Zhang et al., 2012). The retrieval results and relative errors of the mixing states of BC particle groups coated by different coating materials are shown in **Table 2**. For the thinly coated black carbon particles represented by the CCM, both the RR and the RE of mixing states for BC particle groups with different coating materials are well coincident, and the absolute values of RE decrease with the reduction of fractal dimension and preset $D_{p,v}/D_{c,v}$. When $D_p$=2.60, the absolute values of RE for BrC-coated BC decrease from -53.51% to 44.19% as $D_{p,v}/D_{c,v}$ decreases from 2.71 to 2.15. However, for the thickly coated black carbon particles represented by the CAM, the RR and RE of mixing states for BC groups coated by BrC significantly differ from those of the BC particles coated by the other two materials. For BC particles thickly coated by BrC, the retrieval error in mixing state ranges from -53.51% to 6.51%, while the RRs for BC coated by OC or sulfate range from -6.05% to -39.85%. The differences in the RR between BC coated by OC and sulfate are all less than 5.58%. The relative errors in retrieved mixing states of black carbon groups thickly coated by sulfate and OC are smaller than the thinly coated particles. In this study, all the missed points of the retrieved $D_p/D_c$ are taken into account when the RR and RE for particle groups are analyzed, because such coated BC particles with corresponding preset $D_{p,v}/D_{c,v}$ may exist in the atmosphere, but the SP2 indeed misses these particles based on the oversimplified assumption of both morphology and refractive index. Furthermore, the relative errors of the retrieved mixing state of black carbon particles thickly coated by BrC fluctuate positively and negatively, which is also caused by the higher number of missed data points of the retrieval results for BrC-coated BC.

### 3.2 Impact of retrieval errors in mixing states on the estimation of absorption enhancement

During the aging process of BC in the atmosphere, the non-BC materials coat the surface of BC and form inhomogeneous mixing states. The light absorption capacity of BC particles would be enlarged due to the lensing effect, which is described through absorption enhancement ($E_{ab}$) (Cui et al., 2016). As an important optical property, the $E_{ab}$ of the atmospheric coated BC was estimated by Zhang et al. (2023) based on the mixing states measured through field observations using the SP2. However, since the retrieved mixing states of coated BC contain evident retrieval errors, the further use of mixing states for $E_{ab}$ estimation would also lead to inherent errors, which should be quantified. The effect of the SP2 retrieval error of the mixing state on the estimation of enhanced absorption of BC coated by different materials is analyzed in this section from the perspective of numerical simulation.

**Figure 5.** shows the absorption enhancement of the closed-cell model and the core-shell model coated with different coating materials. The mixing state of the core-shell model at each $D_{c,v}$ is set to the retrieved $D_p/D_c$ for the CCM based on the SP2, thus the performance of the SP2-characterized mixing state in the evaluation of $E_{ab}$ can be explored. It can be seen from the

figure that the absorption enhancements of the core-shell models coated by different materials are very similar under all the volume fractions of BC, the values of $E_{ab}$ decrease from about 3.16 to 1.00 as the increase of volume equivalent diameter of the BC core. This is because the SP2 obtains almost the same mixing states for BC coated by all three components, since the fractal BC also has comparable differential scattering properties. As for fractal BC aerosol, the absorption enhancement is not sensitive to the variation of the diameter of the BC core but is sensitive to the coating materials. The actual absorption enhancements of the fractal BC particles coated by sulfate and OC are similar, and the values vary between 1.00 and 2.00 under different volume fractions and diameters of the BC core. However, the absorption enhancements of BC particles coated by BrC significantly differ from those of BC particles coated with the other two materials, because the inherent light absorption of BrC also enlarges the absorption of coated BC. The $E_{ab}$ of BrC-coated BC particles decreases with the decrease of preset volume fraction of BC, resulting in the reduction of the differences in $E_{ab}$ between the BC coated by BrC and the BC coated by the other two components.

In addition, the distinctions in absorption enhancement between thinly coated BC and the corresponding core-shell models are further analyzed. For BC particles coated by BrC, the values of $E_{ab}$ for fractal particles are significantly larger than those for core-shell models when preset $D_{p,v}/D_{c,v}$ ranges from 2.15 to 2.71. In the most extreme case, the $E_{ab}$ for a fractal particle could be about 6.19 times larger than the core-shell model for coated BC with $D_{p,v}/D_{c,v}$=2.71 and BC core diameter of 436 nm. When the mixing states measured by the SP2 are used for the calculation of $E_{ab}$, the absorption enhancement would be significantly underestimated for volume fractions of BC smaller than 0.10. On the contrary, the $E_{ab}$ would be overestimated at first and be underestimated then for BrC-coated BC with $D_{p,v}/D_{c,v}$=1.36, and the turning point corresponds BC diameter of about 350 nm. Similarly, the $E_{ab}$ for fractal BC particles coated by OC and sulfate also have the "overestimation and underestimation" variation compared to the $E_{ab}$ of the core-shell model, but the BC diameter of the turning points varies from about 340 nm to 470 nm, depending on both coating components and BC volume fractions. When $D_{p,v}/D_{c,v}$=1.36, the $E_{ab}$, estimated based on the SP2 retrieved mixing states, for most of the BC particles coated by OC and sulfate are underestimated. In short, the application of mixing states measured by the SP2 could lead to remarkable errors in the estimated $E_{ab}$ of thinly coated BC.

**Figure 6** shows the absorption enhancement of the coated-aggregate model and the core-shell model coated by different materials. The retrieved $D_p/D_c$ for each CAM based on the SP2 is set as the mixing state of the corresponding core-shell model. There are similar absorption enhancements for BC particles represented using core-shell models coated by OC and sulfate under almost all the fractal dimensions and volume fractions of BC, while the absorption enhancements for core-shell models with BrC coatings are larger than those for the other two coatings. The order of the values of $E_{ab}$ is highly correlated with the retrieved results of the mixing states, as shown in Figure 3. The $E_{ab}$ for BrC-coated BC aerosols is significantly larger than the other two coatings, mostly because of the optical absorption of BrC. There are also distinctions in $E_{ab}$ between OC-coated and sulfate-coated BC, which may be due to the compact structure enlarging the effect of the slight difference in refractive indices on the absorption properties. Like the $E_{ab}$ for the thinly coated CCM with $D_{p,v}/D_{c,v}$ larger than 2.15, the $E_{ab}$ for the CAM with BrC also obviously bigger than that of the corresponding core-shell models, which indicates that the retrieval errors in BC mixing state based on the SP2 have an important effect on the evaluation of $E_{ab}$ for BC thickly coated by BrC. For BrC-coated

particles with $D_{p,v}/D_{c,v}$=2.71 and $D_f$=2.60, the $E_{ab}$ for the CAM could reach up to about 3.14 times larger than that of the core-shell model. The values of $E_{ab}$ for BC thickly coated by OC and sulfate, represented using the CAM, show evident fluctuations, and they are comparable to the $E_{ab}$ for the core-shell models. However, the $E_{ab}$ for the CAM is also overestimated at first and then underestimated under most preset $D_{p,v}/D_{c,v}$, just like the CCM. Overall, the evaluation of $E_{ab}$ based on the SP2 measured mixing states would bring obvious errors for BC coated by the three kinds of coatings under almost all the fractal dimensions

and volume fractions of BC, which emphasizes the importance of considering both fractal morphology and coating composition in the optical retrieval principle of the SP2.

### 3.3 Effects of retrieval errors in mixing states on the calculation of BC radiative forcing

When the mixing states of BC characterized by the SP2 are used to further evaluate the radiative effect, the errors in mixing states would result in errors in the optical properties of coated BC, which are essential input parameters of radiative transfer

models. Therefore, these errors in optical properties would be finally transferred to the radiative effects of coated BC aerosols. To assess the inaccuracy of the radiative effects caused by the retrieval errors in BC mixing states, the simple forcing efficiency equation, which describes of the radiative effect of particles on the atmosphere, is used to quantify the radiative forcing of BC particle groups coated by three components at 1064 nm (Peng et al., 2022).

**Table 3** shows the simple forcing efficiencies and their relative errors for BC particle groups coated by sulfate, OC, and BrC.

The radiative effects of the closed-cell and the core-shell model (CSM) of coated BC are compared. The former is calculated using the optical properties such as mass scattering cross-section and mass absorption cross-section of fractal BC particles with preset mixing states, while the latter is calculated using the optical properties of the core-shell model with the retrieved mixing states in the retrieval framework of the SP2. The SFE for the CCM varies inconspicuously with the variation of fractal dimensions under different coatings and BC volume fractions, which means that the fractal shape of coated BC has little effect

on the radiative forcing. However, both the coating components and the preset $D_{p,v}/D_{c,v}$ have obvious effects on the forcing efficiencies of fractal coated BC. As for the core-shell model, their SFEs are not sensitive to either $D_{p,v}/D_{c,v}$, or coatings. Accordingly, the relative errors change with these two factors to different degrees. The relative errors in the simple radiative forcing efficiency of BC particle groups coated by BrC are greater than those of BC coated by the other two materials, and the errors range from -88.73% to -80.50% under different fractal dimensions and mixing states. The relative errors in the simple

radiative forcing efficiency of BC particles coated by sulfate and OC are similar, ranging from -65.22% to -37.50%. That is, when the mixing states characterized by the SP2 are used to evaluate the radiative effect of coated BC, the evaluated value of radiative forcing could be smaller than the inherent values of thinly coated fractal BC particles. In addition, as for thickly coated BC with different materials represented by CAS, the relative errors in simple forcing efficiencies are much more complicated, because of the phenomenon of missing data points and the notable estimation errors in absorption properties of

coated BC, as can be seen in Figure 6. The evaluated radiative forcings of thickly coated BC based on the retrieved $D_{p,v}/D_{c,v}$ can be overestimated or underestimated compared with the actual values, and the evaluated SFE is several times larger than the SFE of fractal BC aerosol. In short, the noteworthy relative errors in the simple forcing efficiencies, which are transferred

from the errors in the mixing state, also demonstrate the necessity of considering both morphologies and coating components in the retrieval principle of the SP2.

## 4. Conclusions

In this study, the mixing state ($D_p/D_c$) of black carbon particles (BC) coated by sulfate, BrC, and OC under various morphological parameters is retrieved to explore the accuracy of the SP2 in measuring the $D_p/D_c$ of coated BC. Two typical morphological models, closed-cell model (CCM) and coated-aggregate model (CAM), are employed to represent thinly and thickly coated BC. The optical properties are calculated based on multiple sphere T-matrix (MSTM) and regarded as optical properties of atmospheric BC, and the optical equivalent $D_p/D_c$ is retrieved based on Mie theory. The retrieval errors in the $D_p/D_c$ are analyzed through the comparisons between the retrieved values and the preset values during the morphological model construction. Furthermore, the effects of the retrieval error in the mixing state on the estimation of absorption enhancement and radiative forcing of BC with different coating materials are analyzed. The main conclusions are summarized as follows:

(1) For thinly coated BC particles, the retrieved $D_p/D_c$ for BC coated by sulfate, BrC, and OC are similar, and the retrieved $D_p/D_c$ for most coated BC particles are underestimated, with the largest error reaching about -63.16%. For thickly coated BC particles, the retrieved $D_p/D_c$ for BrC-coated BC is higher than that for the other two coatings, with the largest error reaching about -63.04%. Some of the retrieved $D_p/D_c$ are missed because of the inherent differences between the fractal BC and the core-shell model, and the missed points of $D_p/D_c$ for BC particles thickly coated by BrC are more numerous than BC coated by the other two materials.

(2) When the diameter of the BC core follows a certain lognormal distribution, the retrieved $D_p/D_c$ and retrieval errors for thinly coated BC gradually decrease with the reduction of the fractal dimension and volume fraction of BC. Typically, the values of relative errors (REs) in the retrieved $D_p/D_c$ decrease from -53.51% to -28.84%. The retrieved $D_p/D_c$ and retrieval errors for BC thickly coated by sulfate and OC are smaller than those of thinly coated BC, while the BC thickly coated by BrC has more fluctuating RE due to the missed point of the retrieved $D_p/D_c$.

(3) The effect of retrieval errors in the mixing state on the estimation of absorption enhancement ($E_{ab}$) of coated BC is analyzed. The $E_{ab}$ of BrC-coated BC particles are larger than the other two coatings in most cases due to the inherent light absorption of the BrC. The largest $E_{ab}$ for thinly coated BC could be about 6.19 times larger than that of the core-shell model. The estimated $E_{ab}$ for fractal BC thinly coated by OC and sulfate is overestimated when the $D_c$ is small, and then underestimated with the turning point of $D_c$ varies from about 340 nm to 470 nm. There are fluctuations in the $E_{ab}$ of fractal BC thickly coated by OC and sulfate with obvious estimation errors.

(4) The retrieved $D_p/D_c$ is used to calculate mass scattering cross-sections and mass absorption cross-sections, and further employed to evaluate the radiative effects at 1064 nm wavelength of coated BC groups. The radiative forcing for coated BC varies with both the coating component and the volume fraction of BC core. The relative errors in the radiative effect of BC

coated by BrC range from -88.73% to -80.00%, while the relative errors for BC coated by sulfate and OC are smaller and range from -65.22% to -37.50%.

Based on the analyses of the errors in the retrieved mixing states and the errors in both the evaluated absorption enhancements and the simple forcing efficiencies, it should be emphasized that both the oversimplification of the morphology of coated BC to the core-shell model and the complex refractive index of coatings to $1.50+0i$ in the measurement framework of the SP2 are

inappropriate. Therefore, the fractal morphologies for both thinly and thickly coated BC and the diversity of coating components should be considered in the retrieval principle of the SP2 to improve the retrieval accuracy and facilitate the wider application of the observed mixing states of BC aerosols. For the future, the more advanced morphological models for coated BC, such as fractal BC with irregular coatings, partially coated BC, and BC with non-spherical monomers, and more diverse coating components, should be considered to further quantify the retrieval errors in the mixing states based on the current

retrieval scheme of the SP2.

*Data availability.* The data for this study is available online (https://doi.org/10.13140/RG.2.2.10735.14249).

*Author contributions.* JL: conceptualization, methodology, funding acquisition, and co-writing of the original draft; DZ:

methodology, experiment, formal analysis, and co-writing of the original draft; GW: experiment, data curation, and manuscript review and editing; CZ: formal analysis, data curation, and manuscript review and editing; and XZ: data curation as well as manuscript review and editing.

## Appendix A: List of Symbols

| | |
|---|---|
| $D_p$ | Optical equivalent diameter of the whole coated BC particle |
| $D_c$ | Volume equivalent diameter of the BC core |
| $D_p/D_c$ | Diameter ratio of the whole particles to the BC core, also the mixing state of coated BC |
| $E_{ab}$ | Absorption enhancement |
| $RF$ | Radiative forcing |
| $a_0$ | Radius of monomer |
| $N_s$ | Number of monomers in the aggregate |
| $k_f$ | Pre-factor |
| $D_f$ | Fractal dimension |
| $R_g$ | Gyration radius |
| $V_f$ | Volume fraction of BC core |
| $D_{p,v}/D_{c,v}$ | Preset mixing state |
| $dC_{sca\text{-MSTM}}$ | Differential scattering cross-section of the fractal BC model |

| $dC_{sca\text{-Mie}}$ | Differential scattering cross-section of the core-shell model |
| $RE$ | Relative error |
| $C_{abs,p}$ | Absorption cross-section of the whole coated BC particle |
| $C_{abs,c}$ | Absorption cross-section of the BC core |
| SFE | Simple forcing efficiency |
| $MSC$ | Mass scattering cross-section |
| $MAC$ | Mass absorption cross-section |
| $\rho_{BC}$ | Density of the BC |
| $\rho_{coating}$ | Density of the coating |
| $RR$ | Retrieval result of the mixing state for the coated BC particle group |
| $d_g$ | Geometric mean diameter |
| $\sigma_g$ | Standard deviation |

***Competing interests.*** The contact author has declared that none of the authors has any competing interests.

***Acknowledgments.*** We particularly thank Dr. Mishchenko M. I. and Dr. Mackowski D. W. for the MSTM code.

***Financial support.*** This research has been supported by the National Natural Science Foundation of China (grant no. 42305082), the China Postdoctoral Science Foundation (grant no. 2024M750493), and the China Meteorological Administration Xiong'an Atmospheric Boundary Layer Key Laboratory (grant no. 2023LABL-B15).

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

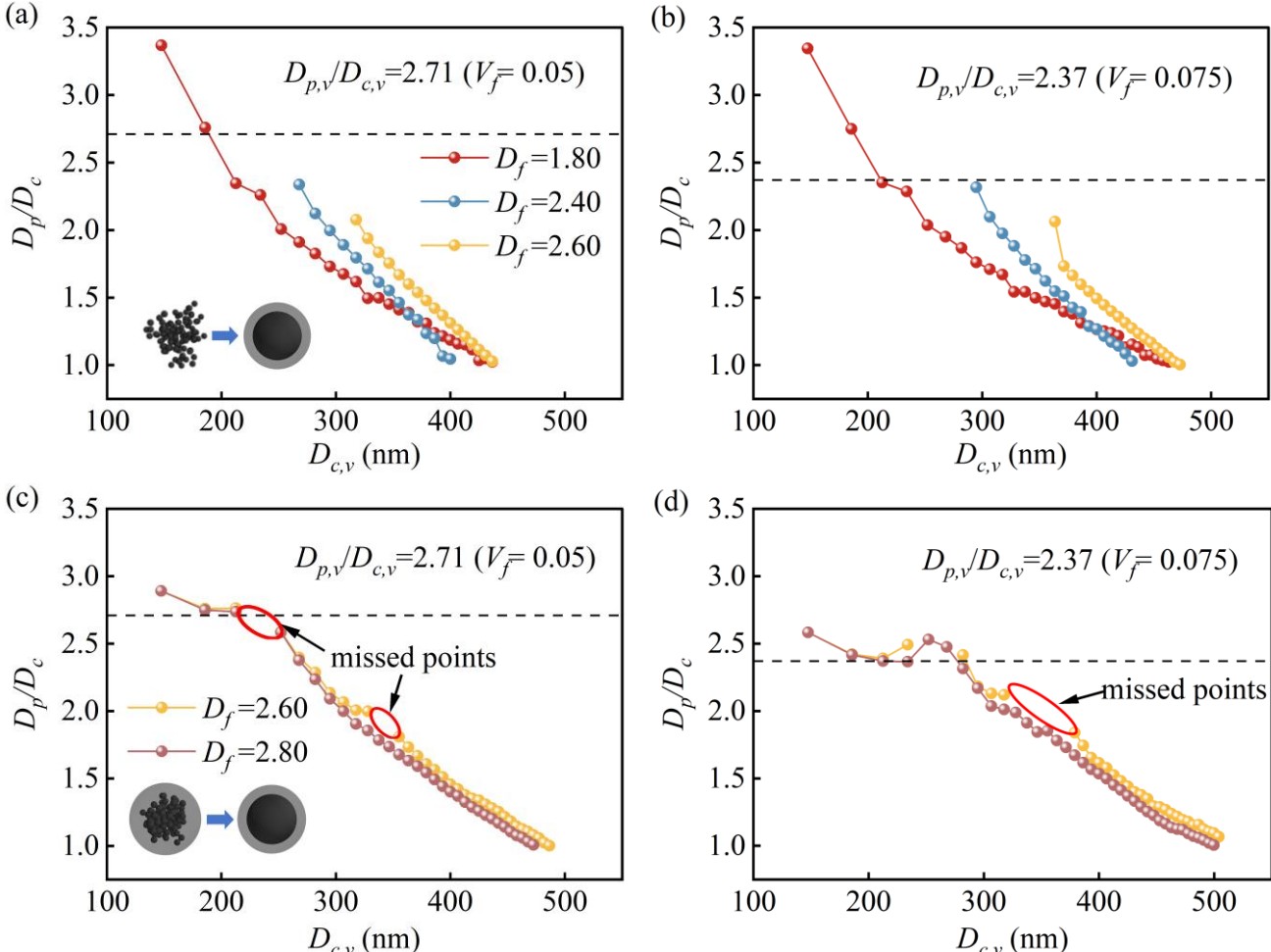

**Figure 1.** The retrieved mixing state ($D_p/D_c$) of sulfate-coated BC particles with different fractal dimensions. (a, b) thinly coated BC particles represented by the closed-cell model with the preset mixing state $D_{p,v}/D_{c,v}$=2.37 and 2.71. (c, d) thickly coated BC particles represented by the coated-aggregate model with the mixing state preset $D_{p,v}/D_{c,v}$=2.37 and 2.71.

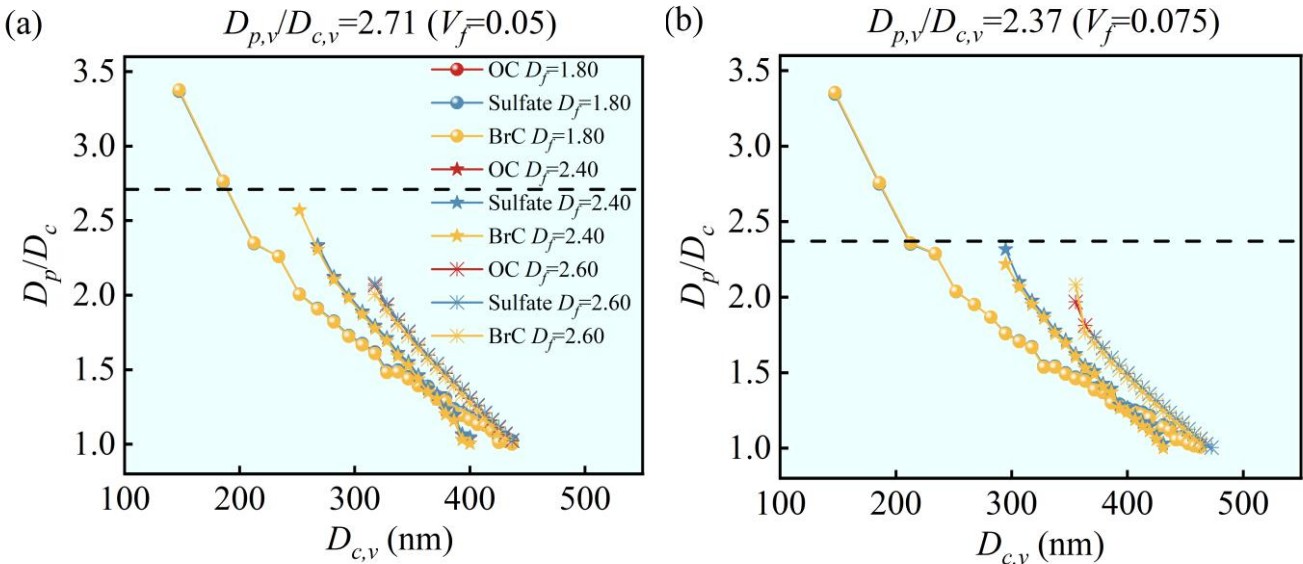

545

**Figure 2**. Retrieved mixing state ($D_p/D_c$) of BC particles coated by different materials represented using the closed-cell model.

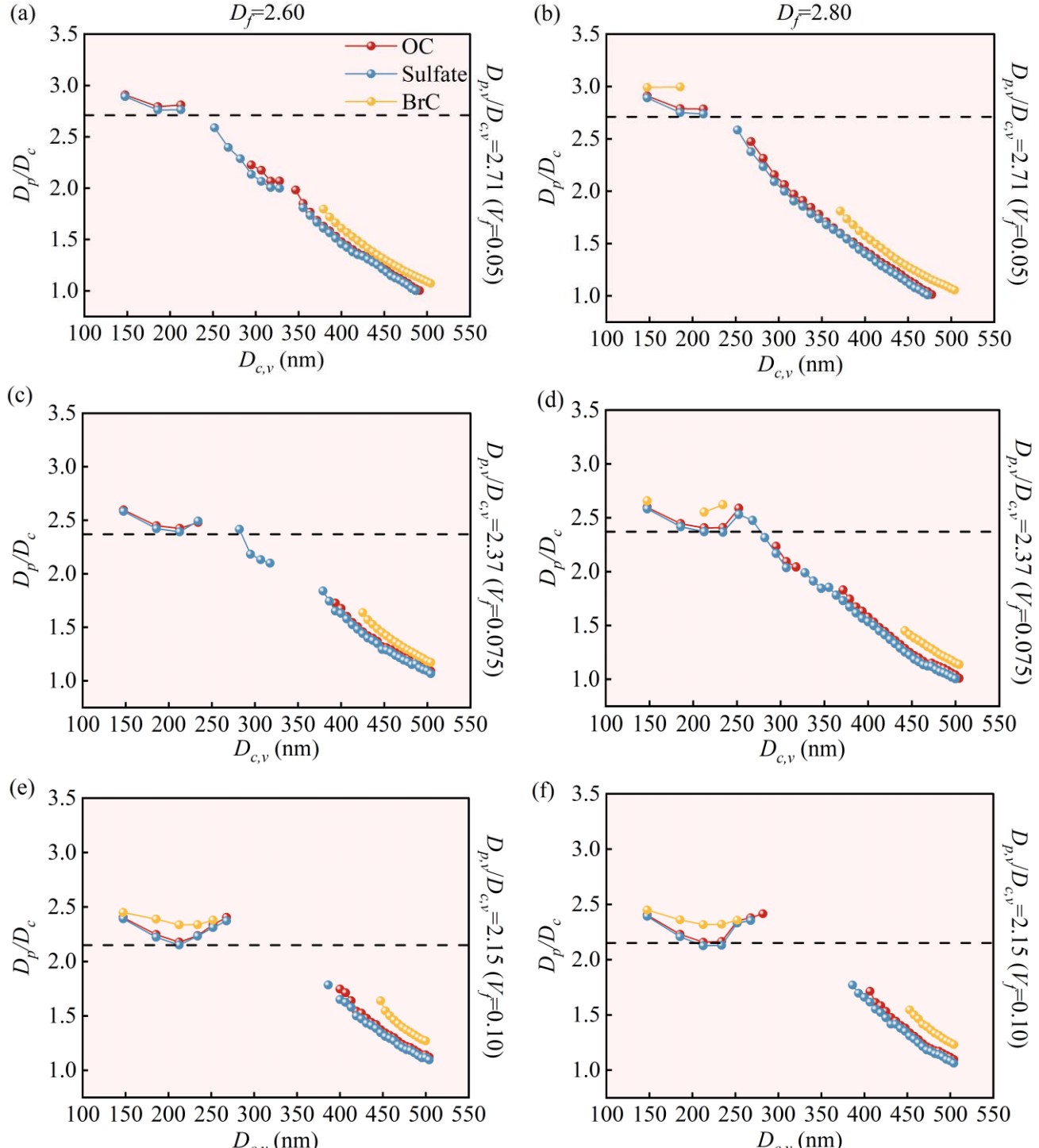

**Figure 3.** Retrieved mixing state ($D_p/D_c$) of BC particles coated by different materials represented using the coated-aggregate model.

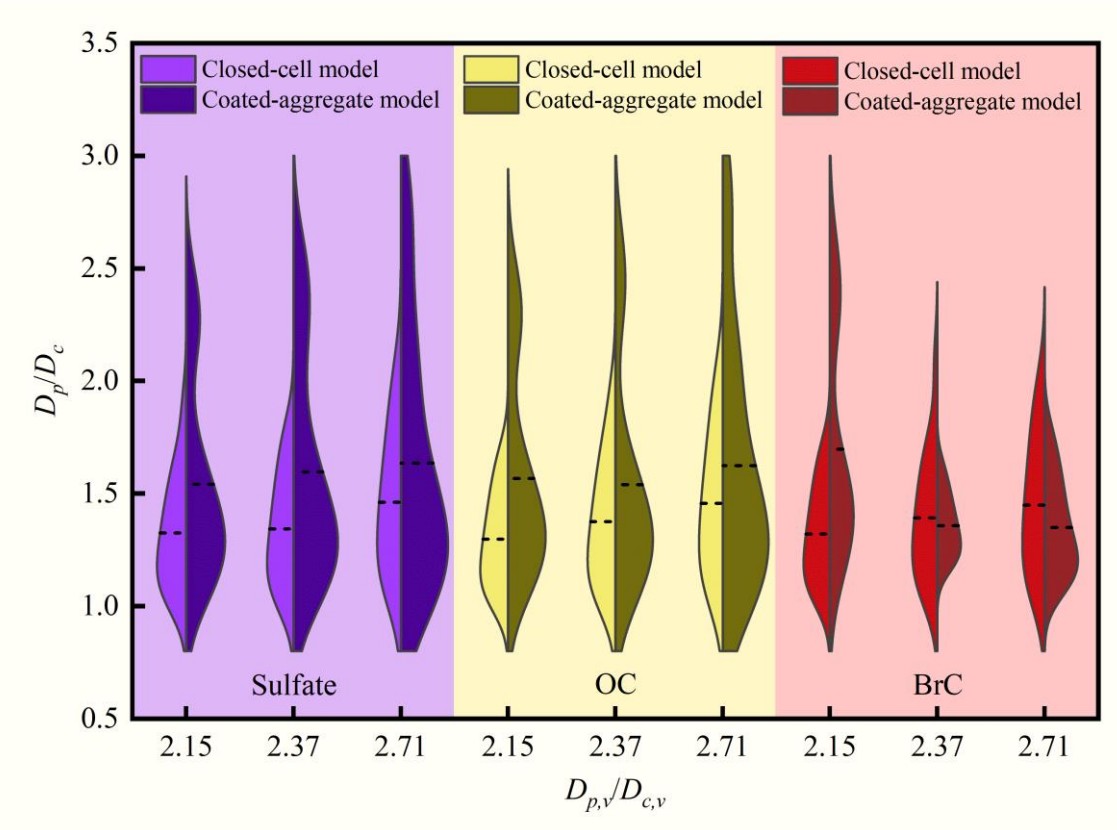

**Figure 4**. Distribution of retrieved mixing states for black carbon particles coated by different materials. The light-colored part on the left of the half-violin diagram is the distribution for the closed-cell model, and the dark-colored part on the right is the distribution for the coated-aggregate model. The width of the half-side filled in the half-violin plot represents the probability distribution of the retrieval results, and the dashed lines represent the average of the retrieval results.

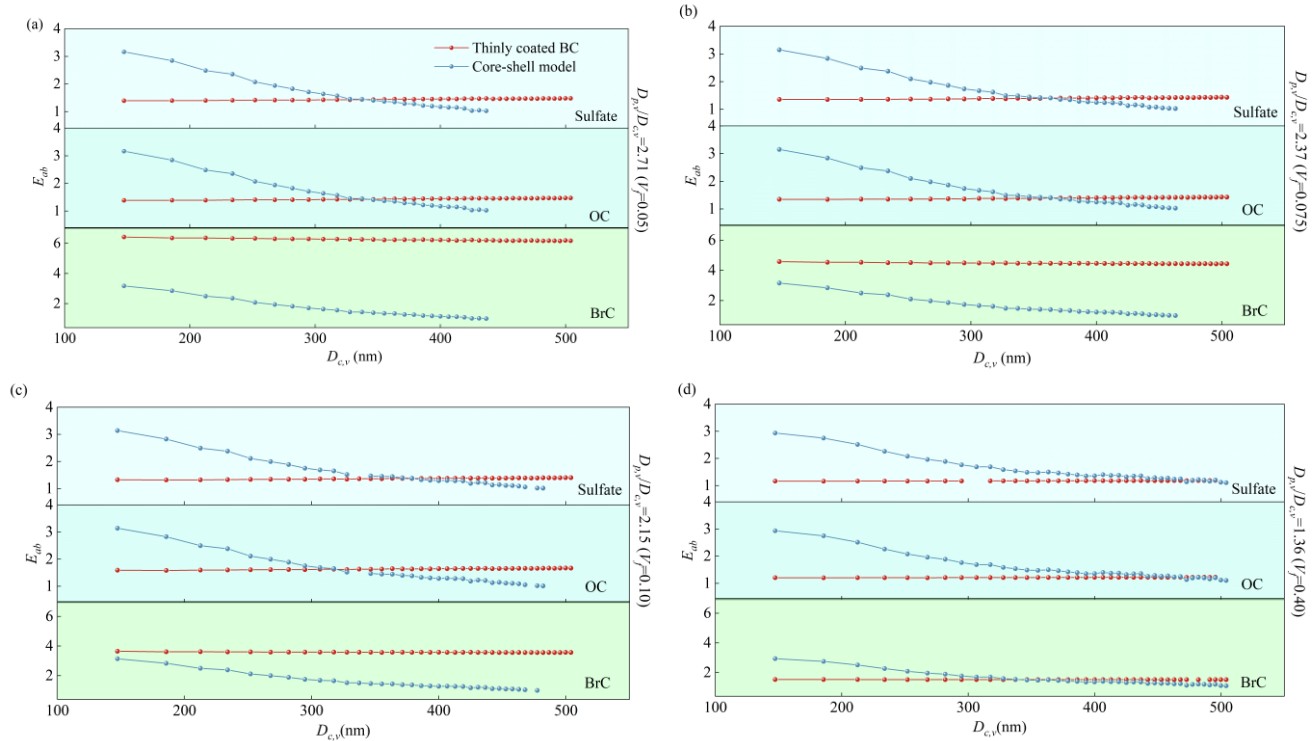

**Figure 5**. The absorption enhancement of both the BC thinly coated by different materials with $D_f = 1.80$ and the core-shell model. (a) The preset $D_{p,v}/D_{c,v}$ is 2.71; (b) The preset $D_{p,v}/D_{c,v}$ is 2.37; (c) The preset $D_{p,v}/D_{c,v}$ is 2.15; (d) The preset $D_{p,v}/D_{c,v}$ is 1.36.

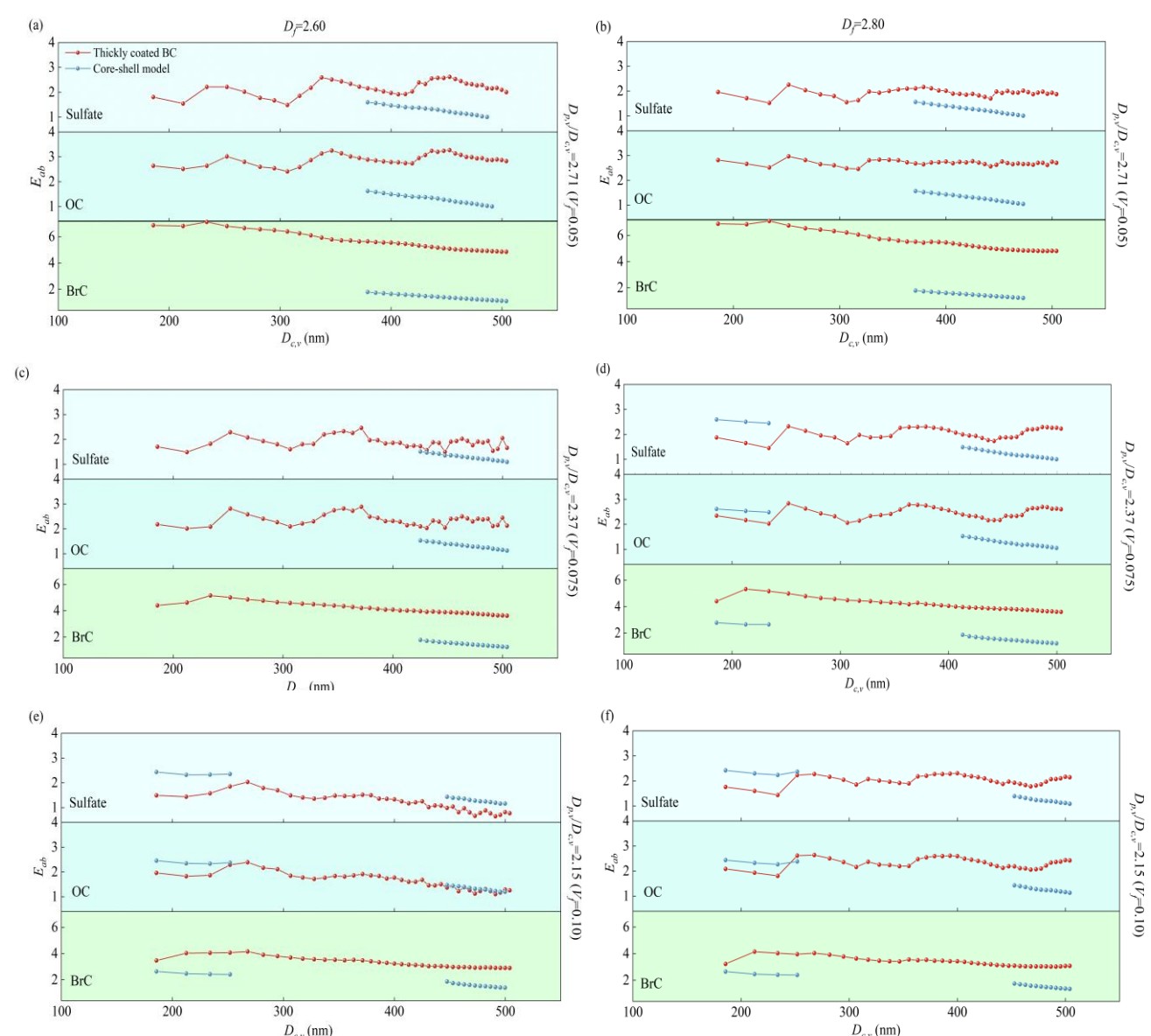

**Figure 6**. The absorption enhancement of both the BC thickly coated by different materials and the core-shell model. (a) The preset $D_{p,v}/D_{c,v}$ is 2.71 and $D_f = 2.60$ ; (b) The preset $D_{p,v}/D_{c,v}$ is 2.71 and $D_f = 2.80$; (c) The preset $D_{p,v}/D_{c,v}$ is 2.37 and $D_f = 2.60$; (d) The preset $D_{p,v}/D_{c,v}$ is 2.37 and $D_f = 2.80$; (e) The preset $D_{p,v}/D_{c,v}$ is 2.15 and $D_f = 2.60$; (f) The preset $D_{p,v}/D_{c,v}$ is 2.15 and $D_f = 2.80$.

**Table 1.** Morphological parameters of coated BC aerosol models.

| Model | Parameters | Values |
|---|---|---|
| All | Monomer radius ($a_0$) | 20 (nm) |
| All | Monomer numbers ($N_s$) | 50-2000, step length 50 |
| All | Fractal prefactor ($k_f$) | 1.20 |
| Closed-cell | Fractal dimension ($D_f$) | 1.80, 2.40, 2.60 |
| | Volume fraction ($V_f$) | 0.70, 0.40, 0.10, 0.075, 0.05 |
| | Volume equivalent diameter ratio of shell/core ($D_{p,v}/D_{c,v}$) | 1.13, 1.36, 2.15, 2.37, 2.71 |
| Coated-aggregate | Fractal dimension ($D_f$) | 2.60, 2.80 |
| | Volume fraction ($V_f$) | 0.10, 0.075, 0.05 |
| | Volume equivalent diameter ratio of shell/core ($D_{p,v}/D_{c,v}$) | 2.15, 2.37, 2.71 |

**Table 2.** Retrieval results and relative errors of the mixing state for black carbon particle groups with different coating materials.

| Model | | Closed-cell model | | | | | | Coated-aggregate model | | | |
|---|---|---|---|---|---|---|---|---|---|---|---|
| $D_f$ | | 1.80 | | 2.40 | | 2.60 | | 2.60 | | 2.80 | |
| $D_{p,v}/D_{c,v}$ | Coating material | RR[a] | RE[b] | RR[a] | RE[b] | RR[a] | RE[b] | RR[a] | RE[b] | RR[a] | RE[b] |
| 2.71 | BrC | 1.45 | -46.49% | 1.27 | -53.14% | 1.26 | -53.51% | 1.26 | -53.51% | 1.64 | -39.48% |
| | Sulfate | 1.49 | -45.02% | 1.31 | -51.66% | 1.26 | -53.51% | 1.65 | -39.11% | 1.62 | -40.22% |
| | OC | 1.49 | -45.02% | 1.30 | -52.03% | 1.25 | -53.87% | 1.63 | -39.85% | 1.63 | -39.85% |
| | BrC | 1.47 | -37.97% | 1.25 | -47.26% | 1.23 | -48.10% | 1.31 | -44.73% | 2.09 | -11.81% |
| 2.37 | Sulfate | 1.51 | -36.29% | 1.28 | -45.99% | 1.20 | -49.37% | 1.91 | -19.41% | 1.75 | -26.16% |
| | OC | 1.51 | -36.29% | 1.28 | -45.99% | 1.20 | -49.37% | 1.96 | -17.30% | 1.78 | -24.89% |
| | BrC | 1.53 | -28.84% | 1.25 | -41.86% | 1.20 | -44.19% | 2.29 | 6.51% | 2.27 | 5.58% |
| 2.15 | Sulfate | 1.53 | -28.84% | 1.21 | -43.72% | 1.19 | -44.65% | 1.97 | -8.37% | 1.93 | -10.23% |
| | OC | 1.53 | -28.84% | 1.21 | -43.72% | 1.18 | -45.12% | 2.02 | -6.05% | 1.99 | -7.44% |

a: The retrieval results of the mixing states for BC particle groups. b: Relative errors in the retrieved mixing states.

**Table 3.** The simple forcing efficiency of BC particle groups coated by different components based on the preset and retrieved mixing states.

| $D_f$ | | 1.80 | | | 2.40 | | | 2.60 | | |
|---|---|---|---|---|---|---|---|---|---|---|
| $D_{p,v}/D_{c,v}$ | Coating material | CCM[a] | CSM[b] | Errors | CCM | CSM | Errors | CCM | CSM | Errors |
| | BrC | 0.71 | 0.08 | -88.73% | 0.71 | 0.09 | -87.32% | 0.71 | 0.08 | -88.73% |
| 2.71 | Sulfate | 0.15 | 0.08 | -46.67% | 0.16 | 0.10 | -37.50% | 0.16 | 0.08 | -50.00% |
| | OC | 0.15 | 0.08 | -46.67% | 0.22 | 0.09 | -59.09% | 0.23 | 0.08 | -65.22% |
| | BrC | 0.51 | 0.07 | -86.27% | 0.51 | 0.08 | -84.31% | 0.51 | 0.08 | -84.31% |
| 2.37 | Sulfate | 0.15 | 0.08 | -46.67% | 0.15 | 0.08 | -44.67% | 0.15 | 0.07 | -53.33% |
| | OC | 0.15 | 0.08 | -46.67% | 0.15 | 0.08 | -44.67% | 0.15 | 0.07 | -53.33% |
| | BrC | 0.40 | 0.07 | -82.50% | 0.40 | 0.08 | -80.00% | 0.41 | 0.07 | -82.93% |
| 2.15 | Sulfate | 0.15 | 0.07 | -53.33% | 0.15 | 0.08 | -46.67% | 0.15 | 0.07 | -53.33% |
| | OC | 0.18 | 0.07 | -61.11% | 0.18 | 0.08 | -55.56% | 0.15 | 0.07 | -53.33% |

a: The closed-cell model. b: The core-shell model.