# Peer review of "Numerical quantitation on the effect of coating materials on the mixing state retrieval accuracy of fractal black carbon based on single particle soot photometer"

_EGUsphere, 2024_

## Referee Comment (RC1)

Review of "Numerical quantitation on the effect of coating materials on the mixing state retrieval accuracy of fractal black carbon based on single particle soot photometer" by Liu et al.

Summary:
In this paper, the authors perform several modeling experiments to evaluate the ways in which different coating materials are likely missed by the DMT SP2 and their effects on retrieved coating diameters, absorption enhancement, and radiative forcing. They find that organic carbon and sulfate coating lead to larger relative errors of the mixing state than black carbon coated with brown carbon and this effect leads to moderate-to-large impacts on the radiative forcing. This work is interesting and provides insightful context on the limitations and potential errors of using SP2 measurements to assess black carbon effects on climate and is well motivated, but the paper is not written well to make these point compelling based on the results. Many details are missing, and the discussion is vague without specific directed attention to the very detailed figures and tables. A large driver of this lack of clarity is the poor writing. The authors should make a concerted effort to carefully review and revise the syntax and diction of this paper so that thoughts are clearly and fully described. I do not recommend this paper for publication in its current form but encourage the authors to make use of the comments and suggestions provided below to improve the paper. Because the issues were somewhat numerous, I was not able to comment on every single one but have highlighted a number of them that I believe can be applied in general.

Major Comments:
- The introduction of this paper is not written well. Many grammatical errors and sentence structure issues persist throughout. On my first pass of the introduction, I started to provide some comments and suggestions (in the Technical Corrections below), but they became far too numerous and minute. The current state of the introduction makes it hard for the reader to follow the motivation of this work. I would suggest taking the specific suggestions provided below and apply them generally throughout this section.

- I am not convinced on the fidelity of this work based on the writing of the Methodology section. Many citations are missing, and the writing is unclear. Several components of this section are not defined and there are far too many variables and parameters to keep track of. I have provided a few suggestions (many issues exist here than I have time and space to note) below that could help with the clarity of this section, but the authors should take more time and care in revising this section so that it can be clear to readers what was done and how.

- The results sections of this work is difficult to follow and lacks specificity. I believe there is a lot (of useful) information provided in each figure and table, but they are not fully described and detailed in the text when they key findings are discussed. I would recommend to the authors to, in addition to providing better figure detail, place their results in context by citing previous literature.

- I believe the conclusion section of this work is inadequate. A vague listing of very general results is not appropriate. More detail, specifically quantified errors in summary, should be provided to what is already included in the text of this section. This section would also benefit from describing the implications of the results that are found in this work. Given the rather large error in radiative forcing, what does this imply for BC effects on climate that motivate how SP2 measurements are used and how they can be improved in the future?

Technical Corrections:
- Abstract: This whole section needs to be revised for clarity and flow. There are several run-on sentences here, missing or redundant commas, and missing articles ("the" before nouns such as, "the SP2"). I will not provide guidance line-for-line, but an example place of improvement includes lines 22-24 which have 3 separate thoughts: (1) the absorption enhancement retrieval deviations due to coatings, (2) the radiative forcing error from coating of BrC, and (3) RF error from sulfate and OC. It is possible that these three thoughts can be combined in one sentence, but the current verbiage is not coherent and far too long of a sentence.

- Line 36: I suggest adding "and" in the sentence: "melting of snow and ice, [and] alters the atmosphere…"

- Lines 37-41: The two sentences in these lines contain disjointed thoughts and do not connect ideas. These should be improved for clarity.

- Lines 42-43, Lines 45-47: These sentences joined by commas should be broken up into two sentences. Appropriate citations for claims should also be added where appropriate.

- Line 49: If the authors are referring to, as I assume, the DMT SP2, they should identify the manufacturer (Droplet Measurement Technologies) and provide a citation for this instrument.

- Line 53: an example of where an article is missing: "The optical cavity of [the] SP2…"

- Line 95: "According to the observations using electron microscopes, …" What observations? Are these the authors' observations or from literature? Please clarify and provide citations where appropriate. The sentence also contains disjointed thoughts; please revise.

- Section 2.1-2.2: Can the authors please provide a table detailing the parameter settings for the models/numerical simulations and provide citations where appropriate. The listing of these parameters in the text alone are very difficult to follow

and keep track of. A table or list of the variables defined in this work (likely as an appendix) should also be provided as there are many variables provided in this section.

- Line 134: MSTM is not defined in the main text before it is used here. Please specify this acronym. In addition, the MSTM algorithm that you used is not detailed anywhere in the methodology section. Did you use a previously written algorithm or did you develop your own. This should be clarified.

- Line 165-167: is there a citation for ASTM G173-03 so that readers can reference where these parameter numbers come from?

- Line 198: clarify "leakage points"

- Discussion of Figure 1 in the main text: The authors should direct the reader to which panel they are discussing when they refer to results shown in the plots.

- Figures 1-3: The authors should quantify the differences in the lines of these plots. Throughout the results section, the differences are discussed as "identical", "similar", "smaller", but these differences are not convincing based on these vague qualities.

- Line 216: If the authors are going to discuss retrieval errors, the table listing these errors should be referenced.

- Tables 1 and 2: please provide in the table caption or footer, a description of the acronyms in the table (RR, RE, etc...)

- Section 3.2: The authors need to write out the numbers that are referenced in Table 1 when they describe the differences. Lots of vague language about what is shown in this section. "The relative error of black carbon particle swarm retrieval results..." what does that mean? If the authors are going to use verbiage such as "significantly different" then a statistical test of difference needs to be done and described.

- Section 3.3: It is unclear to me if a lognormal BC particle distribution was employed for comparison (lines 263-279). If so, how was the lognormal distribution characterized (number concentration, diameter, width) and did you test the sensitivity of these selections?

- Section 3.4: The authors should speak to the atmospheric/climatic implications of the retrieval errors between the SP2 and their model results based on the coating and fractal shape.

- Conclusion: Except for point (4), points (1)-(3) should have quantified differences included. Language such as "similar", "swarm", "small", "different" is not appropriate to describe the differences. The authors need to include numbers to detail what was observed. It also preferred that implications are described based on your results. These should be placed in context with previous literature and what these new results bring to the field.

---

## Referee Comment (RC2)

**Review of "Numerical quantitation on the effect of coating materials on the mixing state retrieval accuracy of fractal black carbon based on single particle soot photometer"**

**General comments**

In this study, the authors used closed-cell model (CCM) and coated-aggregate model (CAM) to retrieve three types of coatings, i.e., sulfate, non-absorbing OC and BrC on black carbon (BC) and investigate their potential impact on the optical properties of BC. It is good that the retrieval uncertainties are also quantified and discussed. The study underscored the importance to consider the coating material when measuring the mixing state of BC and estimating the absorption enhancement (Eab) and radiative forcing (RF). The potential contribution of this study is within the scope of AMT. However, the current version lacks some descriptions of important details and further discussions. It is behind publication quality. Therefore, I recommend that substantial revision needs to be done before considering publication.

**Major comments:**

Definitions of some terms, i.e., Dc, Dp, and mixing state, are not clear in the introduction which causes confusing at the very beginning. However, some of them are very well described in the method section. Please consider reorganizing.

The effect of complex morphology is not very well discussed. Although it is mentioned in the abstract, introduction and methodology, it is not emphasized in the results and discussions. Please consider reshape the manuscript accordingly.

The results section is like experiment report. This should be revised by adding logical linkage between different paragraphs, rather than only listing descriptions of the figures one by one. The authors should reshape this part.

A big concern is that some results are apparently affected by the leakage points; however, the author didn't give enough explanation for such missing data, which needs to be discussed, and an improvement should be considered and/or done. Would it be possible that some of the similar results between sulfate and OC coatings, and their difference from the BrC coating, mainly due to the amount of data points? If yes, this is technical issue which must be well addressed before discussing the scientific questions and drawing conclusions.

The authors used two models and showed the differences between them. As a technical paper, this is good but not enough. It would be necessary to discuss the similarities and differences between their results, and model limitations, respectively.

The conclusion part needs a rewrite after evaluating and/or resolving the issue of leakage points. In addition, the conclusion is not only repeating the results and summarizing the main findings but also discussing the main limitations and the importance of this study in the field of atmospheric science and /or broader field, and proposing future work if possible, etc.

**Minor Comments:**

Line 50: The definition of Dc is unclear here. Please revise. How does the Dp/Dc represent mixing state? Please explain it here. Please define the mixing state in this study. Otherwise, refer the readers to the method section.

Line 57 and 62: Please remove the full names of Dc and Dp but just keep the abbreviations. Please do it throughout the manuscript.

Line 63-65: The current descriptions for light and heavy coatings are unclear. Please add more.

Line 189: Define real and imaginary components of refractive index, and give references.

Line 202: Please give the reasons for that SP2 will lose data in the retrieval of the mixing state of BC with BrC coating.

Line 204-206: Remove the figure caption, which should only be shown with the corresponding figure. Please do it for all the figure and table captions embedded in the main text.

Line 212: This statement is not accurate: not all the retravel results of BrC coating are obviously different from the other two coatings. Only 1/3 of the results, i.e., Fig 2 e) and f) show the obvious difference, which is huge and interesting. Please revise and give the explanation.

Fig 1: c) and d) why there are some data gaps, especially for Df=2.6? What are the differences between light and heavy coatings? d) Change Y scale to keep consistency with the other three.

Fig 2: Keep consistency of Y and X scales; change color codes to better distinguish sulfate and BrC.

Fig 3: Keep consistency of Y and X scales; It is hard to compare Fig 2 and 3 due to the criteria chosen for showing the results. Please try to keep consistency and if it cannot be maintained, please explain the reasons and/or limitations.

Fig 6: The legend is the same as Fig 5 rather than Fig 4. Please revise.

---

## Author Comment (AC1)

**RESPONSE LETTER OF REVISION (EGUSPHERE-2024-3781)**

Title: Numerical quantitation on the effect of coating materials on the mixing state retrieval accuracy of fractal black carbon based on single particle soot photometer

Dear Editor and Referee:

We have revised our manuscript based on the comments. The corrections and modifications have been included in the revised manuscript and the details are listed as follows. The responses are highlighted in blue font. The changes made in the revised manuscript are marked in red font.

**Summary:**

In this paper, the authors perform several modeling experiments to evaluate the ways in which different coating materials are likely missed by the DMT SP2 and their effects on retrieved coating diameters, absorption enhancement, and radiative forcing. They find that organic carbon and sulfate coating lead to larger relative errors of the mixing state than black carbon coated with brown carbon and this effect leads to moderate-to-large impacts on the radiative forcing. This work is interesting and provides insightful context on the limitations and potential errors of using SP2 measurements to assess black carbon effects on climate and is well motivated, but the paper is not written well to make these point compelling based on the results. Many details are missing, and the discussion is vague without specific directed attention to the very detailed figures and tables. A large driver of this lack of clarity is the poor writing. The authors should make a concerted effort to carefully review and revise the syntax and diction of this paper so that thoughts are clearly and fully described. I do not recommend this paper for publication in its current form but encourage the authors to make use of the comments and suggestions provided below to improve the paper. Because the issues were somewhat numerous, I was not able to comment on every single one but have highlighted a number of them that I believe can be applied in general.

Response:

Thanks a lot for reviewing our manuscript and the constructive comments. All

these comments and suggestions are of great significance for improving the quality of our manuscript. We have reorganized and rewritten most of the manuscript. More detailed analyses have been added in the revised manuscript, and the syntax and diction are carefully revised. We have responded to all the comments point by point, and related descriptions have been modified accordingly in the revised manuscript.

**Major Comments:**

• The introduction of this paper is not written well. Many grammatical errors and sentence structure issues persist throughout. On my first pass of the introduction, I started to provide some comments and suggestions (in the Technical Corrections below), but they became far too numerous and minute. The current state of the introduction makes it hard for the reader to follow the motivation of this work. I would suggest taking the specific suggestions provided below and apply them generally throughout this section.

Response:

Thank you for the valuable comments. We have reorganized the Introduction section to improve the overall logical coherence and make the motivation clearer. The grammatical errors and sentence structure issues have also been corrected to our best. The detailed modifications are shown in the revised manuscript.

I am not convinced on the fidelity of this work based on the writing of the Methodology section. Many citations are missing, and the writing is unclear. Several components of this section are not defined and there are far too many variables and parameters to keep track of. I have provided a few suggestions (many issues exist here than I have time and space to note) below that could help with the clarity of this section, but the authors should take more time and care in revising this section so that it can be clear to readers what was done and how.

Response:

Thank you very much for the suggestions. Most descriptions in the Methodology section have been modified. All the variables and parameters have been clearly defined, and necessary references about the parameters and their values have been added and cited. The whole section has also been reorganized to provide much clearer descriptions about model construction for coated BC, optical property simulation, retrieval process of the mixing state, and the evaluation of both absorption enhancement and radiative effect. The detailed modifications are given in the revised manuscript.

The results sections of this work is difficult to follow and lacks specificity. I believe there is a lot (of useful) information provided in each figure and table, but they are not fully described and detailed in the text when they key findings are discussed. I would recommend to the authors to, in addition to providing better figure detail, place their results in context by citing previous literature.

Response:

Thanks for your valuable comments. The "Results and discussion" section has been reorganized and rewritten in the revised manuscript. More information in the Figures and Tables is described and analyzed in detail. The influences of coating structures, morphological parameters, and coating materials on the retrieved mixing states of coated BC are discussed. The effects of the retrieval errors in retrieved mixing states on the evaluation accuracies of absorption enhancement and simple forcing efficiency are also analyzed. In addition, important references have been added and cited as needed. The detailed modifications can be found in the revised manuscript.

I believe the conclusion section of this work is inadequate. A vague listing of very general results is not appropriate. More detail, specifically quantified errors in summary, should be provided to what is already included in the text of this section. This section would also benefit from describing the implications of the results that are found in this

work. Given the rather large error in radiative forcing, what does this imply for BC effects on climate that motivate how SP2 measurements are used and how they can be improved in the future?

Response:

Thanks for your valuable suggestion. We have modified the conclusion section in the revised manuscript, with more descriptions about quantified errors in the retrieved mixing states as well as the evaluated absorption enhancements and radiative effects. Based on the results of the study, we found that the assumption of the core-shell model for coated BC and the assumption of $1.50+0i$ for the refractive index of the coating in the retrieval scheme of the SP2 are inappropriate. The mixing states of coated BC measured by the SP2 would contain remarkable errors. When the measurement results are further employed to evaluate the radiative forcing of BC aerosol, the evaluated results would also be inaccurate. Therefore, the fractal morphologies for coated BC and the diversity of coatings should be considered in the retrieval principle of the SP2, then the accuracy of the retrieved mixing state and the evaluated radiative effect of BC could be improved. In addition, more advanced morphological models and more kinds of coating materials should be considered for further quantification of retrieval errors caused by inappropriate assumptions. Detailed modifications are given in the revised manuscript.

**Technical Corrections:**

Abstract: This whole section needs to be revised for clarity and flow. There are several run-on sentences here, missing or redundant commas, and missing articles ("the" before nouns such as, "the SP2"). I will not provide guidance line-for-line, but an example place of improvement includes lines 22-24 which have 3 separate thoughts: (1) the absorption enhancement retrieval deviations due to coatings, (2) the radiative forcing error from coating of BrC, and (3) RF error from sulfate and OC. It is possible that these three thoughts can be combined in one sentence, but the current verbiage is not coherent and far too long of a sentence.

Response:

Thank you for the constructive suggestions. We have reviewed and rewritten most of the Abstract in the revised manuscript. The sentence in lines 22-24 has been split for clarity. The modifications in the revised manuscript are as follows:

"Fractal closed-cell model and coated-aggregate model are constructed to represent BC with thin and thick coatings, multiple sphere T-matrix (MSTM) simulation results are regarded as optical properties of realistic coated BC, and the mixing states are optically retrieved based on Mie theory according to the retrieval principle of the SP2."

"Furthermore, when the SP2 measured mixing states are employed to estimate the absorption enhancement ($E_{ab}$) and radiative forcing of BC aerosols, the estimation error of $E_{ab}$ for BC with BrC coating is significantly larger than that with sulfate and OC coatings, and the estimation $E_{ab}$ could be 6.19 times of the inherent value of fractal BC. Compared with fractal BC, the radiative forcing estimation error at 1064 nm for BrC-coated BC can reach -88.7%, while the estimation errors of BC coated by the other two materials range from -64.3% to -38.4%. This study highlights that the coating materials and fractal morphology should be considered when the SP2 is used to measure the mixing state of BC."

Line 36: I suggest adding "and" in the sentence: "melting of snow and ice, [and] alters the atmosphere…"

Response:

Thanks for your comments. The related descriptions have been modified in the revised manuscript as follows:

"BC also prevents the vertical diffusion of pollutants by altering the atmospheric temperature structure and accelerates the melting of snow and ice when it deposits on their surfaces (Ding et al., 2016)."

Lines 37-41: The two sentences in these lines contain disjointed thoughts and do not connect ideas. These should be improved for clarity.

Response:

Thank you for the comments. We have modified these two sentences in Lines 37-41 in the revised manuscript as follows:

"Optical absorption-dominated fresh BC particles usually present as loose chain-like aggregates consisting of numerous near-spherical monomers (Ceolato et al., 2020). During the aging in the atmosphere, fresh BC particles are coated by various chemical components such as sulfate, non-absorbing organic carbon (OC), and brown carbon (BrC) through the condensation and coagulation processes, forming aged BC aerosols with complex mixing structures (China et al., 2013). The heterogeneous mixing has a complicated and profound influence on the radiative effect of BC aerosols."

Lines 42-43, Lines 45-47: These sentences joined by commas should be broken up into two sentences. Appropriate citations for claims should also be added where appropriate.

Response:

Thanks a lot for your comments. The descriptions in Lines 42-43 have been modified, and necessary references have been added. The descriptions in Lines 45-47 were removed to improve the logical coherence of the whole paragraph. The modifications in the revised manuscript are as follows:

"The "mixing state" is a key microphysical property for aged BC to describe the mixing structure of BC and coating, and it can be characterized through different principles and instruments (Liu et al., 2023b; Liu et al., 2025)."

Line 49: If the authors are referring to, as I assume, the DMT SP2, they should identify the manufacturer (Droplet Measurement Technologies) and provide a citation for this instrument.

Response:

Thank you very much for the suggestions! We have modified related descriptions and added references in the revised manuscript as follows:

"The single particle soot photometer (SP2, Droplet Measurement Technology, Inc.) is one of the effective online instruments that can effectively measure the mixing state of coated BC, based on the combination of laser-induced incandescence and light scattering measurement, and has been widely employed in laboratory and field observations (Liu et al., 2020a; Liu et al., 2022; Schwarz, 2019)."

Line 53: an example of where an article is missing: "The optical cavity of [the] SP2…"
Response:

Thanks for your valuable comments! We have reviewed the whole manuscript. The articles have been added as needed in the revised manuscript.

Line 95: "According to the observations using electron microscopes, …" What observations? Are these the authors' observations or from literature? Please clarify and provide citations where appropriate. The sentence also contains disjointed thoughts; please revise.
Response:

Thank you for the helpful comments. According to the electron microscope observation results of China et al. (2013), Liu et al. (2020), and Wang et al. (2021), two fractal models of were constructed in this study. For simplification, the related descriptions have been modified, and references have been added in the revised manuscript as follows:

"The freshly emitted bare BC particles are fractal clusters that consist of a large number of near-spherical monomers, following the scaling rule (China et al., 2013; Liu et al., 2020b; Wang et al., 2021b):"

Section 2.1-2.2: Can the authors please provide a table detailing the parameter settings for the models/numerical simulations and provide citations where appropriate. The listing of these parameters in the text alone are very difficult to follow and keep track of. A table or list of the variables defined in this work (likely as an appendix) should also be provided as there are many variables provided in this section.

Response:

Thank you for the valuable comments. We have added a table to summarize the morphological parameters of BC models in the revised manuscript. In section 2.2, there are parameters and variables related to optical simulation, microphysical parameters, and radiative evaluation. This section has been rewritten to improve clarity and logicality. According to the order of optical property simulation, mixing state retrieval, and absorption enhancement and radiative effect calculation, some important equations are given in sequence. The variables in the equations and supplementary formulas are defined, and the corresponding value settings are given with necessary references. Furthermore, a list of the variables in the study is provided as an appendix. More detailed modifications have been made in the revised manuscript as follows:

"Table 1. Morphological parameters of coated BC aerosol models."

| Model | Parameters | Values |
|---|---|---|
| All | Monomer radius ($a_0$) | 20 (nm) |
| All | Monomer numbers ($N_s$) | 50-2000, step length 50 |
| All | Fractal prefactor ($k_f$) | 1.20 |
| Closed-cell | Fractal dimension ($D_f$) | 1.80, 2.40, 2.60 |
| | Volume fraction ($V_f$) | 0.70, 0.40, 0.10, 0.075, 0.05 |
| | Volume equivalent diameter ratio of shell/core ($D_{p,v}/D_{c,v}$) | 1.13, 1.36, 2.15, 2.37, 2.71 |
| Coated-aggregate | Fractal dimension ($D_f$) | 2.60, 2.80 |
| | Volume fraction ($V_f$) | 0.10, 0.075, 0.05 |
| | Volume equivalent diameter ratio of shell/core ($D_{p,v}/D_{c,v}$) | 2.15, 2.37, 2.71 |

Line 134: MSTM is not defined in the main text before it is used here. Please specify this acronym. In addition, the MSTM algorithm that you used is not detailed anywhere in the methodology section. Did you use a previously written algorithm or did you develop your own. This should be clarified.

Response:

Thanks a lot for your comments. The MSTM employed in this study is the acronym of the multiple sphere $T$-matrix algorithm developed by Mackowski and Mishchenko (2011). The related descriptions have been modified in the revised manuscript as follows:

"In this study, optical properties such as optical cross-section and scattering matrix at 1064 nm, which is consistent with the observation wavelength of the SP2, are simulated using the multiple sphere T-matrix (MSTM) method developed by Mackowski and Mishchenko (2011)."

Line 165-167: is there a citation for ASTM G173-03 so that readers can reference where these parameter numbers come from?

Response:

Thank you very much for the comments! We have added and cited necessary references for the parameter settings in the equation of simple forcing efficiency in the revised manuscript as follows:

"The $dS(\lambda)/d\lambda$ is the spectral solar irradiance according to ASTM G173-03 (ASTM G173-03 Reference Spectra https://www.nrel.gov/grid/solar-resource/spectra.html), the atmospheric transmittance $\tau$=0.79 (Bhandari et al., 2019), the cloud fraction $F_c$=0.6 (Bond and Bergstrom, 2006), the typical urban surface albedo $a_s$=0.19 (Liu et al., 2023b), and the backscattering fraction $\beta$=0.15 (Peng et al., 2022)."

Line 198: clarify "leakage points"

Response:

Thanks a lot for your helpful comments! To improve clarity, we replaced the "leakage points" with the "missed point". When the differential scattering cross-sections of the fractal BC cannot be matched using the core-shell models based on the retrieval principle of the SP2, there would be missed points in the retrieved mixing states. And the essential reason for the missed points is that the inherent difference in the morphological model results in distinct optical properties. The related descriptions have been modified as follows:

"It should be noted that there are a certain number of missed data points of retrieved $D_p/D_c$, because of the inherent differences between the fractal coated BC model and the spherical core-shell model. More specifically, there is no core-shell model whose differential optical properties could match the coated BC particle with certain parameters corresponding to the missed points."

Discussion of Figure 1 in the main text: The authors should direct the reader to which panel they are discussing when they refer to results shown in the plots.

Response:

Thank you very much for the comments. Specific subfigure numbers have been mentioned in the main text when the corresponding subfigures are discussed and analyzed for all the figures in the manuscript. The descriptions related to Figure 1 have been modified in the revised manuscript as follows:

"As can be seen from Fig.1 (a) and (b) that the retrieved $D_p/D_c$ for thinly coated BC decreases with the diameter of the BC core, and the retrieved mixing states for BC core diameters larger than 200 nm are smaller than the preset values, which signifies that the SP2 will underestimate the mixing states for coated BC whose core size larger than 200 nm."

"Fig. 1 (c) and (d) indicate that the values of *RE* for thinly sulfate-coated BC particles with $D_f$=2.60 represented by the CCM are larger than those of thickly coated BC particles represented by the CAM, especially for larger BC volume fraction, which reveals that the SP2 has better performance in characterizing the mixing state of thickly coated BC."

Figures 1-3: The authors should quantify the differences in the lines of these plots. Throughout the results section, the differences are discussed as "identical", "similar", "smaller", but these differences are not convincing based on these vague qualities.

Response:

Thank you for the valuable suggestions. The descriptions of Figures 1-3 have been rewritten. The results are discussed from both qualitative and quantitative perspectives. Specific values of retrieval errors in the mixing states of coated BC have also been illustrated. Detailed modifications are shown in the revised manuscript.

Line 216: If the authors are going to discuss retrieval errors, the table listing these errors should be referenced.

Response:

Thanks a lot for your comments. There were mistakes in the data usage for the retrieved mixing states of the closed-cell model coated by BrC. Figure 2 has been modified and redrawn in the revised manuscript. The related descriptions in Line 216 have been removed. Quantitative discussions about the retrieval errors in mixing states are added according to necessity in the revised manuscript.

Tables 1 and 2: please provide in the table caption or footer, a description of the acronyms in the table (RR, RE, etc…)

Response:

Thank you very much for pointing this out. We have added footnotes to explain the meanings of the acronyms in the related Tables in the revised manuscript as follows:

"**Table 2.** Retrieval results and relative errors of the mixing state for black carbon particle groups with different coating materials."

| Model | | Closed-cell model | | | | | | Coated-aggregate model | | | |
|---|---|---|---|---|---|---|---|---|---|---|---|
| $D_f$ | | 1.80 | | 2.40 | | 2.60 | | 2.60 | | 2.80 | |
| $D_{p,v}/D_{c,v}$ | Coating material | RRs[a] | REs[b] | RRs | REs | RRs | REs | RRs | REs | RRs | REs |
| | BrC | 1.45 | -46.8% | 1.27 | -53.2% | 1.26 | -53.7% | 1.26 | -53.5% | 1.64 | -39.6% |
| 2.71 | Sulfate | 1.49 | -45.2% | 1.31 | -51.8% | 1.26 | -53.6% | 1.65 | -39.2% | 1.62 | -40.3% |
| | OC | 1.49 | -45.2% | 1.30 | -52.0% | 1.25 | -53.9% | 1.63 | -40.0% | 1.63 | -40.1% |
| | BrC | 1.47 | -37.9% | 1.25 | -47.2% | 1.23 | -48.2% | 1.31 | -44.6% | 2.09 | -12.0% |
| 2.37 | Sulfate | 1.51 | -36.5% | 1.28 | -45.9% | 1.20 | -49.5% | 1.91 | -19.5% | 1.75 | -26.2% |
| | OC | 1.51 | -36.5% | 1.28 | -45.9% | 1.20 | -49.2% | 1.96 | -17.5% | 1.78 | -25.0% |
| | BrC | 1.53 | -29.0% | 1.25 | -42.1% | 1.20 | -44.1% | 2.29 | 6.4% | 2.27 | 5.2% |
| 2.15 | Sulfate | 1.53 | -28.9% | 1.21 | -43.8% | 1.19 | -44.9% | 1.97 | -8.7% | 1.93 | -10.2% |
| | OC | 1.53 | -29.1% | 1.21 | -43.9% | 1.18 | -45.2% | 2.02 | -6.2% | 1.99 | -7.4% |

a: The retrieval results of the mixing states for BC particle groups. b: Relative errors in the retrieved mixing states.

"**Table 3.** The simple forcing efficiency of BC particle groups coated by different components based on the preset and retrieved mixing states."

| $D_f$ | | 1.80 | | | 2.40 | | | 2.60 | | |
|---|---|---|---|---|---|---|---|---|---|---|
| $D_{p,v}/D_{c,v}$ | Coating material | CCM[a] | CSM[b] | Errors | CCM | CSM | Errors | CCM | CSM | Errors |
| | BrC | 0.71 | 0.08 | -88.7% | 0.71 | 0.09 | -87.3% | 0.71 | 0.08 | -88.5% |
| 2.71 | Sulfate | 0.15 | 0.08 | -45.0% | 0.16 | 0.10 | -38.4% | 0.16 | 0.08 | -48.4% |
| | OC | 0.15 | 0.08 | -45.0% | 0.22 | 0.09 | -57.4% | 0.23 | 0.08 | -64.3% |
| | BrC | 0.51 | 0.07 | -85.6% | 0.51 | 0.08 | -84.2% | 0.51 | 0.08 | -85.2% |
| 2.37 | Sulfate | 0.15 | 0.08 | -49.1% | 0.15 | 0.08 | -44.4% | 0.15 | 0.07 | -54.0% |
| | OC | 0.15 | 0.08 | -49.2% | 0.15 | 0.08 | -44.3% | 0.15 | 0.07 | -53.3% |
| | BrC | 0.40 | 0.07 | -83.1% | 0.40 | 0.08 | -80.5% | 0.41 | 0.07 | -82.9% |
| 2.15 | Sulfate | 0.15 | 0.07 | -51.9% | 0.15 | 0.08 | -48.5% | 0.15 | 0.07 | -55.6% |
| | OC | 0.18 | 0.07 | -60.3% | 0.18 | 0.08 | -57.2% | 0.15 | 0.07 | -55.4% |

a: The closed-cell model. b: The core-shell model.

Section 3.2: The authors need to write out the numbers that are referenced in Table 1 when they describe the differences. Lots of vague language about what is shown in this section. "The relative error of black carbon particle swarm retrieval results…" what does that mean? If the authors are going to use verbiage such as "significantly different" then a statistical test of difference needs to be done and described.

Response:

Thanks a lot for your constructive comments. To improve the logical coherence of the discussion, we combined Sections 3.1 and 3.2 in the original manuscript into one subsection. The particle groups have been clarified in the revised manuscript, and the vague language has also been modified. Quantitative comparisons have been analyzed based on the values of retrieval results and retrieval errors for the mixing states of BC coated by different materials. Detailed modifications are shown in the revised manuscript.

Section 3.3: It is unclear to me if a lognormal BC particle distribution was employed for comparison (lines 263-279). If so, how was the lognormal distribution characterized (number concentration, diameter, width) and did you test the sensitivity of these selections?

Response:

Thank you very much for the helpful comments. The typical lognormal distribution for the volume equivalent diameter of the BC core is employed to discuss the retrieved mixing states and the radiative effect for BC aerosol groups with different coating materials. As parameters describing the distribution pattern, the geometric mean diameter is assumed to be 0.15 μm, and the standard deviation is set to 1.59 in this study, which are the parameter values frequently used by optical numerical simulation research (Liu et al., 2023; Yu and Luo, 2009; Zhang et al., 2012). The related descriptions have been modified in the revised manuscript as follows:

"Atmospheric BC aerosols mostly exist as particle groups that follow certain size distributions. The retrieval results (RR) of the mixing states for BC particle groups

coated by different components and the corresponding retrieval errors (RE) are analyzed, under the assumption that the volume equivalent diameter of BC follows the typical lognormal distribution:

$$n(d) = \frac{1}{\sqrt{2\pi} d \ln(\sigma_g)} \exp\left[ -\left( \frac{\ln(d) - \ln(d_g)}{\sqrt{2}\ln(\sigma_g)} \right)^2 \right] \tag{14}$$

where $d$ is the diameter of BC, $d_g$ and $\sigma_g$ are the geometric mean diameter and standard deviation, which are set to 0.15 μm and 1.59 in this study, respectively (Yu and Luo, 2009; Zhang et al., 2012)."

Section 3.4: The authors should speak to the atmospheric/climatic implications of the retrieval errors between the SP2 and their model results based on the coating and fractal shape.

Response:

Thanks a lot for the constructive suggestions. We have added descriptions about the atmospheric implications in the Conclusion section for logical coherence in the revised manuscript as follows:

"Based on the analyses of the errors in the retrieved mixing states and the errors in both the evaluated absorption enhancements and the simple forcing efficiencies, it should be emphasized that the oversimplification of the morphology of coated BC to the core-shell model and the complex refractive index of coatings to 1.50+0$i$ is inappropriate. The mixing states of coated BC measured by the SP2 would contain remarkable errors. When the measurement results are further employed to evaluate the radiative forcing of coated BC, the evaluated results would also be inaccurate. Therefore, the fractal morphologies for both thinly and thickly coated BC and the diversity of coating components should be considered in the retrieval principle of the SP2 to improve the retrieval accuracy and facilitate the wider application of the observation of the mixing states of BC aerosols."

Conclusion: Except for point (4), points (1)-(3) should have quantified differences included. Language such as "similar", "swarm", "small", "different" is not appropriate to describe the differences. The authors need to include numbers to detail what was observed. It also preferred that implications are described based on your results. These should be placed in context with previous literature and what these new results bring to the field.

Response:

Thank you very much for the helpful comments and suggestions. We have rewritten the Conclusion in the revised manuscript. The points (1)-(3) have been reorganized with quantitative analyses, and the vague language has also been modified. The atmospheric implications of this study and future works have been emphasized. Research focusing on the effects of BC structure and coating on measurement errors of the SP2 is still relatively scarce. Our previous study investigated the effect of different morphological parameters on the optical retrieval errors in the mixing state, and the results are in line with the BC coated by sulfate and OC in this current study (Liu et al., 2023). According to the necessities of the discussion, the discoveries in our previous study have been added and cited in Section 3.1 in the revised manuscript.

**Reference**

China, S., Mazzoleni, C., Gorkowski, K., Aiken, A. C., and Dubey, M. K.: Morphology and mixing state of individual freshly emitted wildfire carbonaceous particles, Nature Communications, 4, 7, 10.1038/ncomms3122, 2013.

Liu, J., Zhang, Q., Wang, J., and Zhang, Y.: Light scattering matrix for soot aerosol: Comparisons between experimental measurements and numerical simulations, Journal of Quantitative Spectroscopy and Radiative Transfer, 246, 106946, 2020.

Liu, J., Wang, G., Zhu, C., Zhou, D., and Wang, L.: Numerical investigation on retrieval errors of mixing states of fractal black carbon aerosols using single-particle soot photometer based on Mie scattering and the effects on radiative forcing estimation, Atmos. Meas. Tech., 16, 4961-4974, 10.5194/amt-16-4961-2023, 2023.

Mackowski, D. W. and Mishchenko, M. I.: A multiple sphere T-matrix Fortran code for

use on parallel computer clusters, Journal of Quantitative Spectroscopy and Radiative Transfer, 112, 2182-2192, 2011.

Wang, Y. Y., Li, W. J., Huang, J., Liu, L., Pang, Y. E., He, C. L., Liu, F. S., Liu, D. T., Bi, L., Zhang, X. Y., and Shi, Z. B.: Nonlinear Enhancement of Radiative Absorption by Black Carbon in Response to Particle Mixing Structure, Geophys. Res. Lett., 48, 10, 10.1029/2021gl096437, 2021.

Yu, F. and Luo, G.: Simulation of particle size distribution with a global aerosol model: contribution of nucleation to aerosol and CCN number concentrations, Atmos. Chem. Phys., 9, 7691-7710, 10.5194/acp-9-7691-2009, 2009.

Zhang, K., O'Donnell, D., Kazil, J., Stier, P., Kinne, S., Lohmann, U., Ferrachat, S., Croft, B., Quaas, J., Wan, H., Rast, S., and Feichter, J.: The global aerosol-climate model ECHAM-HAM, version 2: sensitivity to improvements in process representations, Atmos. Chem. Phys., 12, 8911-8949, 10.5194/acp-12-8911-2012, 2012.

---

## Author Comment (AC2)

**RESPONSE LETTER OF REVISION (EGUSPHERE-2024-3781)**

**Title:** Numerical quantitation on the effect of coating materials on the mixing state retrieval accuracy of fractal black carbon based on single particle soot photometer

Dear Editor and Referee:

We have revised our manuscript based on the comments. The corrections and modifications have been included in the revised manuscript and the details are listed as follows. The responses are highlighted in blue font. The changes made in the revised manuscript are marked in red font.

General comments

In this study, the authors used closed-cell model (CCM) and coated-aggregate model (CAM) to retrieve three types of coatings, i.e., sulfate, non-absorbing OC and BrC on black carbon (BC) and investigate their potential impact on the optical properties of BC. It is good that the retrieval uncertainties are also quantified and discussed. The study underscored the importance to consider the coating material when measuring the mixing state of BC and estimating the absorption enhancement ($E_{ab}$) and radiative forcing (RF). The potential contribution of this study is within the scope of AMT. However, the current version lacks some descriptions of important details and further discussions. It is behind publication quality. Therefore, I recommend that substantial revision needs to be done before considering publication.

Response:

Thanks a lot for reviewing our manuscript and for all these constructive comments. All these comments are very helpful in further refining the manuscript, and we have responded to the comments point by point. Necessary details and discussions of results have been supplemented in the revised manuscript. Thank you again for reviewing our manuscript.

**Major comments:**

Definitions of some terms, i.e., Dc, Dp, and mixing state, are not clear in the introduction which causes confusing at the very beginning. However, some of them are very well described in the method section. Please consider reorganizing.

Response:

Thank you very much for the valuable comments and suggestions. We have reorganized the relevant descriptions of the important terms in the revised manuscript. The $D_c$ denotes the equivalent volume diameter of the black carbon core, and $D_p$ denotes the optical equivalent diameter of the coated black carbon. The particle diameter ratio of the whole particles to the BC core ($D_p/D_c$) is used to characterize the mixing state. The related description has been modified in the revised manuscript as follows:

"The differential scattering property can be derived subsequently, and the optical equivalent diameter of the whole coated BC particle ($D_p$) can be retrieved based on the Mie scattering theory in combination with the spherical core-shell model."

"After the coating is completely evaporated, the bare BC emits incandescent signals, since the peak of the incandescent signal is proportional to BC mass, the volume equivalent diameter of BC core ($D_c$) can be obtained by using BC density of 1.8 g/cm$^3$ (Schwarz et al., 2006; Bond and Bergstrom, 2006)."

"Finally, the mixing state of coated BC at the single particle level can be characterized by the SP2 as the particle diameter ratio of the whole particles to the BC core ($D_p/D_c$) (Schwarz et al., 2008a; Schwarz et al., 2008b)."

The effect of complex morphology is not very well discussed. Although it is mentioned in the abstract, introduction and methodology, it is not emphasized in the results and discussions. Please consider reshape the manuscript accordingly.

Response:

Thanks for this constructive comment. We have added in-depth discussions about the effects of complex morphology (coating structures, fractal parameters, and volume fractions of BC core) on the retrieval results of the mixing states and the evaluation of both absorption enhancement and radiative forcing in the revised manuscript.

The results section is like experiment report. This should be revised by adding logical linkage between different paragraphs, rather than only listing descriptions of the figures one by one. The authors should reshape this part.

Response:

Thank you for your valuable comments. We have rewritten most of the section of "result and discussion" in the revised manuscript.

A big concern is that some results are apparently affected by the leakage points; however, the author didn't give enough explanation for such missing data, which needs to be discussed, and an improvement should be considered and/or done. Would it be possible that some of the similar results between sulfate and OC coatings, and their difference from the BrC coating, mainly due to the amount of data points? If yes, this is technical issue which must be well addressed before discussing the scientific questions and drawing conclusions.

Response:

Thanks for the valuable suggestion. The reason for the existence of a certain number of missed points of the retrieved $D_p/D_c$ is the inherent differences between the fractal BC model and the spherical core-shell model, in other words, there is no core-shell model whose differential optical scattering properties could match the coated BC particle with morphological parameters corresponding to the missed points. As for the single particle level, the missed points would not affect the analysis of variation trends of the results. While the missed points do affect the retrieved mixing states and the evaluated radiative effects for particle groups. However, the phenomenon of missed points of the SP2 due to the unreasonable assumptions of both the particle shape and the refractive index of the coating is what we want to stress. And the coating materials have obvious effects on the amount of missed points, the BrC coating leads to more missed points. To truly reflect the missed points of the SP2 and evaluate the effects of the missed points on the further application of the observation results of the SP2, the missed points are all considered in the evaluation of absorption enhancement and radiative forcing in our study.

The authors used two models and showed the differences between them. As a technical paper, this is good but not enough. It would be necessary to discuss the similarities and differences between their results, and model limitations, respectively.

Response:

Thank you for the meaningful comment.

The closed-cell model and the coated aggregate model are constructed to represent thinly coated and thickly coated BC, respectively. Due to the inherent optical difference between the fractal model and the core-shell model, the retrieval results of the mixing states for both kinds of fractal models contain obvious retrieval errors. The effects of the fractal dimension and volume fractions of BC on the retrieved mixing state are similar, while the effects of coating materials are slightly different. The mixing states for most coated BC, represented by two fractal models, would be underestimated by the SP2. Even though there are widespread phenomena of missed points, the amount of missed points is quite distinct for different models with various morphological parameters. When the retrieved mixing states are used to evaluate the absorption enhancement of coated BC, the evaluated $E_{ab}$ is overestimated at first and then underestimated for most cases. In short, when compared with the core-shell model, the closed-cell model and coated aggregate model show some similarity, however, there are also obvious differences between these two models coated by three components.

In addition, even though the closed-cell and coated-aggregate models could represent realistic coated BC to some extent, with the development of the morphology model, there are more advanced models for coated BC, such as fractal BC with more irregular coatings, partially coated BC, and BC with non-spherical monomers (Luo et al., 2018). Therefore, to more precisely evaluate the optical retrieval errors in the mixing states using the SP2, more realistic morphological models and more coating types still need to be employed. Necessary descriptions have been added to the revised manuscript.

"For the future, the more advanced morphological models for coated BC, such as fractal BC with irregular coatings, partially coated BC, and BC with non-spherical monomers, and more diverse coating components should be considered to further

quantify the retrieval errors in the mixing states using the SP2."

The conclusion part needs a rewrite after evaluating and/or resolving the issue of leakage points. In addition, the conclusion is not only repeating the results and summarizing the main findings but also discussing the main limitations and the importance of this study in the field of atmospheric science and /or broader field, and proposing future work if possible, etc.

Response:

Thank you for your valuable comments on our research. We have revised the conclusion section to summarize our main findings in a better way. The obvious retrieval errors in the mixing states measured by the SP2 due to the unsuitable assumption on both the core-shell structure and the single complex refractive index $1.50+0i$ of coatings are stressed. We emphasize the fractal morphologies and the coating components should be considered in the retrieval principle of the SP2 to improve the retrieval accuracy and facilitate the application of the observed mixing states of BC. We also put forward that more advanced morphological models should be considered to obtain more valuable information about the effect of complex shape and diversity of coatings on the measurement accuracy of the mixing state of BC using the SP2. Specific revisions are detailed in the revised manuscript.

**Minor Comments:**

Line 50: The definition of $D_c$ is unclear here. Please revise. How does the $D_p/D_c$ represent mixing state? Please explain it here. Please define the mixing state in this study. Otherwise, refer the readers to the method section.

Response:

Thank you very much for the valuable comments. We have reorganized the relevant descriptions of the important terms in the revised manuscript. The $D_c$ denotes the volume equivalent diameter of the black carbon core, and $D_p$ denotes the optical

equivalent diameter of the coated particle. The diameter ratio of the whole coated particle to the BC core ($D_p/D_c$) is used by the SP2 to characterize the mixing state of coated BC. The related descriptions have been modified in the revised manuscript as follows:

"The differential scattering property can be derived subsequently, and the optical equivalent diameter of the whole coated BC particle ($D_p$) can be retrieved based on the Mie scattering theory in combination with the spherical core-shell model."

"After the coating is completely evaporated, the bare BC emits incandescent signals, since the peak of the incandescent signal is proportional to BC mass, the volume equivalent diameter of BC core ($D_c$) can be obtained by using BC density of 1.8 g/cm$^3$ (Schwarz et al., 2006; Bond and Bergstrom, 2006)."

"Finally, the mixing state of coated BC at the single particle level can be characterized by the SP2 as the particle diameter ratio of the whole particles to the BC core ($D_p/D_c$) (Schwarz et al., 2008a; Schwarz et al., 2008b)."

Line 57 and 62: Please remove the full names of $D_c$ and $D_p$ but just keep the abbreviations. Please do it throughout the manuscript.

Response:

Thank you for the comments. The full names of $D_c$ and $D_p$ involved have been modified to abbreviations in the revised manuscript, except for the first time they appear.

"The morphology simplification of aged BC would result in inherent errors in the $D_p$ and the mixing state."

"Our previous studies have preliminarily revealed that fractal morphology and coating structure can result in mixing state retrieval errors up to approximately 80%, and it is worth noting that the characterization of $D_p/D_c$ based on Mie theory would miss some amount of mixing state results for coated BC with certain microphysical parameters (Liu et al., 2023b)."

"Nevertheless, the refractive index of the coating shell is roughly assumed to be constant 1.50+0$i$ in the current optical retrieval scheme of the SP2, so the single refractive index would also lead to retrieval errors in both $D_p$ and $D_p/D_c$."

Line 63-65: The current descriptions for light and heavy coatings are unclear. Please add more.

Response:

Thanks for your constructive comments. Based on the lag time between the appearance of the scattered signal and the incandescent signal recorded by the SP2, the coated BC can be classified into thinly and thickly coated BC. BC with light coatings means that the BC aggregate is covered by a thin coating film, which can be represented by the closed-cell model. On the contrary, BC with heavy coatings means that the BC aggregate is fully encapsulated in a mass of coating material, which can be represented by the coated aggregate model. The related descriptions in the revised manuscript have been modified as follows:

"More specifically, the BC core and outer coating are assumed to be concentric double spheres, while the realistic BC can be an aggregate covered by a film (thinly-coated) or encapsulated in a package (heavily-coated) (Qin et al., 2022)."

"The CCM is also a fractal aggregate whose monomer is a concentric double-layer sphere, the inner sphere represents BC, and the outer sphere represents the coating. For the construction of the CCM, the original fractal aggregate generated by DLA is enlarged according to the BC volume fraction, and then a soot sphere is added into each enlarged monomer. As for the coated aggregate model (CAM), the whole fractal aggregate generated by DLA is encapsulated using a spherical coating (Liu et al., 2023a)."

Line 189: Define real and imaginary components of refractive index, and give references.

Response:

Thank you very much for the meaningful comments. The real and imaginary parts of the complex refractive index have been defined in the revised manuscript, and the necessary reference has been added as follows:

"The distinction in complex refractive indices of coatings is one of the fundamental reasons for the optical differences. As for coating materials, the real part of the complex refractive index refers to the ratio of the propagation speed of light in a vacuum to that in the coatings, reflecting the scattering ability of the coating material. While the imaginary part refers to the attenuation of light during the propagation, reflecting the absorption ability of the coating material (Mishchenko et al., 2002)."

Line 202: Please give the reasons for that SP2 will lose data in the retrieval of the mixing state of BC with BrC coating.

Response:

Thank you very much for the valuable comments. The SP2 misses data in the retrieved mixing states of BC because of the inherent optical distinction between the realistic fractal BC coated by different materials and the core-shell model with single refractive index of the shell. Furthermore, not only would the BrC-coated BC lose some of the mixing states, but similar phenomena also occur for BC coated by sulfate and OC, for the same reason. More detailed explanation have been added in the revised manuscript:

"It should be noted that there are a certain number of missed data points of retrieved $D_p/D_c$, because of the inherent differences between the fractal coated BC model and the spherical core-shell model. More specifically, there is no core-shell model whose differential optical properties could match the coated BC particle with certain parameters corresponding to the missed points. There are missed values of retrieved $D_p/D_c$ for sulfate coated both CCM and CAM with BC core diameter ranges from about 200 to 400 nm, just like the situation revealed by Liu et al. (2023b)."

Line 204-206: Remove the figure caption, which should only be shown with the corresponding figure. Please do it for all the figure and table captions embedded in the main text.

Response:

    Thank you for the suggestions. All the captions of both figures and tables embedded in the main text have been removed in the revised manuscript.

Line 212: This statement is not accurate: not all the retravel results of BrC coating are obviously different from the other two coatings. Only 1/3 of the results, i.e., Fig 2 e) and f) show the obvious difference, which is huge and interesting. Please revise and give the explanation.

Response:

    Thanks a lot for the valuable comments. After careful examination, we discovered that there were mistakes in the data usage and plotting for the retrieved mixing state of the closed-cell model coated by BrC with $D_f$=2.60. We are very sorry for these mistakes. Figure 2 has been modified and redrawn in the revised manuscript as shown below. The retrieval results of the mixing state ($D_p$/$D_c$) for the closed-cell model coated by different materials are very similar due to the coupling effect of coating structure, volume fraction of BC core, and the relatively small difference in refractive indices for different coating components. The related descriptions and explanations have been modified in the revised manuscript.

    "Essentially, the BC aerosols with different coating materials have almost identical optical properties, including differential scattering properties observed by the SP2, due to the joint influence of refractive index and coating structure. On the one hand, compared with the BC, the distinctions in both the real and imaginary parts of the refractive indices of all these coatings at 1064 nm are not very significant. On the other hand, the soot cores are distributed in each of the coating monomers in the closed-cell model, the interaction of each soot would be weakened due to the isolation of coatings.

Furthermore, the lensing effect of the coating would also be reduced with the volume fraction of BC decreasing to a certain degree, like 0.075 and 0.05 in this study (Lack and Cappa, 2010). The enlargement of optical properties caused by the coating would also be inconspicuous anymore. This is also the reason why the retrieved results of $D_p/D_c$ for coated BC are comparable with the $V_f$ increases from 0.05 to 0.075."

[Figure]

"Figure 2. Retrieved mixing state ($D_p/D_c$) of BC particles coated by different materials represented using the closed-cell model."

Fig 1: c) and d) why there are some data gaps, especially for Df=2.6? What are the differences between light and heavy coatings? d) Change Y scale to keep consistency with the other three.

Response:

Thank you for the constructive comments. The reason for the missed data points of the retrieved $D_p/D_c$ is that there are inherent optical differences for the fractal BC and the core-shell model, so when the differential optical properties of the core-shell model are used to match those of the fractal BC, such a core-shell model whose differential optical properties are similar to the fractal BC cannot be found. The fractal dimension controls the compactness of fractal BC models. When the fractal dimension for coated-aggregate models is 2.80, the shape of the BC core becomes a sphere. The coated BC is close to a core-shell model in morphology, and the optical properties are close to the core-shell model. Thus, the mixing state for coated BC with $D_f$=2.80 can be retrieved to some degree and shows fewer missed points. On the contrary, the optical

distinctions between BC with a smaller fractal dimension ($D_f$=2.60) and the core-shell model are larger, which leads to more missed points.

The thinly coated BC is represented by the closed-cell model (CCM), which is a fractal aggregate formed by core-shell monomers. While the thickly coated BC is represented by the coated-aggregate model (CAM), which consists of a bare BC aggregate and a spherical coating. Even though the retrieved $D_p/D_c$ for both CCM and CAM decrease with the diameter increase of the BC core, the retrieved $D_p/D_c$ for these two models show different sensitivity to the coating structure and the morphological parameters such as fractal dimension and volume fraction of BC core. The variation of morphological parameters also leads to different amounts of missed points and retrieval errors in the retrieved $D_p/D_c$.

Figure 1 has been modified in the revised manuscript, and related descriptions have also been modified.

[Figure]

"**Figure 1.** Retrieved mixing state ($D_p/D_c$) of sulfate-coated BC particles with different fractal dimensions. (a, b) thinly coated BC particles with the preset $D_{p,v}/D_{c,v}$=2.37 and 2.71. (c, d) thickly coated BC particles with the preset $D_{p,v}/D_{c,v}$=2.37 and 2.71."

"With the fractal dimension increasing from 2.60 to 2.80, the BC core becomes a sphere and the coated-BC is closer to the core-shell model, optical distinctions between fractal BC and the core-shell model are smaller, which reduces the missed points of retrieved $D_p/D_c$."

"As can be seen from Fig.1 (a) and (b) that the retrieved $D_p/D_c$ for thinly coated BC decreases with the diameter of the BC core, and the retrieved mixing states for BC core diameters larger than 200 nm are smaller than the preset values, which signifies that the SP2 will underestimate the mixing states for coated BC whose core size larger than 200 nm. On the contrary, when the BC core is smaller than 200 nm, the SP2 will overestimate the mixing states for thinly coated BC. With the increase of the fractal dimension, the fractal closed-cell structure becomes more compact the retrieval errors in $D_p/D_c$ decrease. The increase of BC volume fraction, that is a small amount of coatings, also leads to reduced values of *RE*. Fig. 1 (c) and (d) indicate that the values of *RE* for thinly sulfate-coated BC particles with $D_f$=2.60 represented by the closed-cell model are larger than those of thickly coated BC particles represented by the coated-aggregate model, especially for larger BC volume fraction, which reveals that the SP2 has better performance in characterizing the mixing state of thickly coated BC. Unlike the thinly coated BC, as the increase of $D_f$, the retrieved $D_p/D_c$ for the thickly coated BC deviates more from the preset value, the *RE* becomes more obvious."

Fig 2: Keep consistency of Y and X scales; change color codes to better distinguish sulfate and BrC.

Response:

Thank you very much for the valuable comments. We have redrawn Figure 2 in the revised manuscript for better comparison and understanding. Since the retrieved mixing states for BC particles coated by different materials under various fractal dimensions and volume fractions of BC core are very similar, even different colors and symbols in the figure cannot exhibit these distinctions in the retrieved $D_p/D_c$.

[Figure]

"Figure 2. Retrieved mixing state ($D_p/D_c$) of BC particles coated by different materials represented using the closed-cell model."

Fig 3: Keep consistency of Y and X scales; It is hard to compare Fig 2 and 3 due to the criteria chosen for showing the results. Please try to keep consistency and if it cannot be maintained, please explain the reasons and/or limitations.

Response:

Thank you very much for the suggestions. We have redrawn both Figure 2 and Figure 3 in the revised manuscript to keep the consistency of both Y and X scales.

[Figure]

"**Figure 3**. Retrieved mixing state ($D_p/D_c$) of BC particles coated by different materials represented using the coated-aggregate model."

Fig 6: The legend is the same as Fig 5 rather than Fig 4. Please revise.

Response:

  Thanks a lot for pointing this out. The related descriptions have been modified in the revised manuscript as follows:

  "**Figure 6**. The absorption enhancement of BC thickly coated by different

materials and the corresponding core-shell model. (a) The preset $D_{p,v}/D_{c,v}$ is 2.71 and $D_f = 2.60$ ; (b) The preset $D_{p,v}/D_{c,v}$ is 2.71 and $D_f = 2.80$ ; (c) The preset $D_{p,v}/D_{c,v}$ is 2.37 and $D_f = 2.60$; (d) The preset $D_{p,v}/D_{c,v}$ is 2.37 and and $D_f = 2.80$; (e) The preset $D_{p,v}/D_{c,v}$ is 2.15 and $D_f = 2.60$; (f) The preset $D_{p,v}/D_{c,v}$ is 2.15 and and $D_f = 2.80$."

**Reference**

Luo, J., Zhang, Y. M., Wang, F., and Zhang, Q. X.: Effects of brown coatings on the absorption enhancement of black carbon: a numerical investigation, Atmos. Chem. Phys., 18, 16897-16914, 10.5194/acp-18-16897-2018, 2018.

---

## Referee Report (RR1)

**General comments:**

This paper analyses the effect of the simplified hypothesis underlying the SP2 data processing used to determine the mixing state of black carbon. In particular, the authors tested the impact of using different coating materials with different refractive indices (sulphates, organic carbon and brown carbon) and of giving the coated BC particles different shapes (ranging from more fractal-like to a core-shell model) on the mixing state. The mixing state is defined as the diameter ratio between the BC core and the whole BC particle, and is calculated from SP2 measurements using the usual retrieval method. This method uses Mie theory and assumes a core-shell morphology and a fixed refractive index. The authors identified biases resulting from these simplifications, particularly for thinly coated BC. BC coated with BrC also exhibited higher inaccuracies due to its absorbing properties. The authors further investigated subsequent errors in radiative forcing and absorption enhancement. Since the SP2 is the only online instrument that can measure the coating thickness of BC-containing particles at a particle-by-particle level, the methodology to process its mixing state data is crucial for interpreting the results This study provides valuable insights into errors in the BC mixing state resulting from core-shell and fixed refractive index assumptions. Furthermore, the results suggest that a more accurate characterisation of BC optical and radiative properties would be achieved by considering more complex morphologies and variable refractive indices. The tests and experiments conducted in this study are robust and well described. However, the good scientific quality of the paper is still suffering from a significant number of grammatical errors. Some of them are listed bellow. I would strongly recommend that the authors carefully review the entire manuscript in terms of English writing, before publication.

**Specific comments:**

l 14. I would add particles : coated black carbon  particles (BC)

l 14. "optical properties"

l. 16. "the coating material"

l. 16 "diverse"

l. 21. "are used to study optical properties…"

l. 27 "be up to 6.15 times higher than"

l 33-34 This sentence is not clear since human = anthropogenic. Please reformulate.

L 36 "optical absorption properties"

l. 39 "BC particles are usually"

l. 40 "BC particles can be coated by various"

l. 42: "through condensation and coagulation"

l. 47 " one of the most effective online instrument that can measure"

l. 57 It would be useful to precise that the change in the Gaussian pattern is due to the different refractive indices between the coating and the core.

l. 58 : Please provide references for the refractive indices mentioned here.

l.59 : I would start a new sentence from the word "since" to add clarity.

l.60 "by assuming spherical particles and using BC density …"

l.64 "into bare-to-thinly coated"

l. 69 The work package is a bit weird here, please reformulate

l 74. "sulfate or organics"

l. 79 "the measurements of the SP2 are usually…"

l. 83 " are built to represent"

l. 84 "can be classified using the lag time"

l. 100" is the gyration radius"

l. 100 formatting of the ith

l. 109 Please don't start a sentence with "and"

l. 112 "are built" . That would be nice to add a schematic of the typical shapes of these two different models.

l. 135 "For the quantification"

Formula 6 : Please define what is $X^2$. More generally, it would be better to introduce each before its corresponding definition (equations 9, 10, 11,12,13), as the authors did for i.e. equation 7.

L 150-151 This sentence is not grammatically correct, or something is missing maybe ?

L 152 "the precision in the estimated typical optical"

l. 163-164 Keep the same structure, either first variable and then definition as the beginning of the sentence or the other way around

l. 176 "the variation of complex refractive"

l. 178-180 The sentence starting with "while" is not grammatically correct.

l. 184-187 The sentence starting with "as can be" is not grammatically correct.

l.189 "more compact and the retrieval"

l. 189 "BC volume fraction due to a smaller amount"

l. 192 "SP2 has a better performance"

l.193 . This sentence is not clear to me. Please reformulate.

l. 198 "from about 200 to 400 nm, as shown by Liu et al."

l.201 "missing data points of retrieved"

l.202-203 Not only that but it will also lead to a bias toward size ranges of coated BC that the SP2 is able to characterize when considering the ensemble of BC particle.

Figure 1 : This is good to include a schematic to show the corresponding morphologies of BC but I don't really understand what values of Dc/Dp and Dc,v each of the representation refers to. Maybe move the images more to the corners, ie. Maximum value of Dc/Dp?

Figure 3 : The Dp,v/Dc,v in vertical on the right hand side is a bit difficult to understand. Maybe move to the left, add a space between it and the y-label, and increase the font? Same for the column titles, increase font and space between title and figures

Figure 4 : Describe what is the dashed line on the violins

Caption of Figure 5 : "…and the corresponding core-shell model, , represented for different values of Dp,v/Dc,v."

Caption of Figure 6: Same as previous comment

l. 207 "to the coating components, thus the"

l. 211 move the "compared with the BC refractive index" to the end of the sentence

l. 212 "on the other hand, when the soot"

l. 215 Do you mean Cappa study, by "this study" ? Then citation needs to be done properly : Cappa et al found …

l. 215 : The sentence starting with "The enlargement" is not clear to me. Please reformulate.

l. 220 This was already said l193, so maybe change to " that confirms..."

l. 230 "between sulfate and OC could explain their similar Dp/Dc"

l. 231 : If "could" is used for the first part of the sentence, then one would use "may have caused" or so

l. 240 A higher accuracy than for other coating materials ?

l. 242 "The coupling effect […] would lose" : not grammatically correct. Would lead to …

l. 247 "Three distinct regions coloured in"

l. 248-250 : It would be nice to put that in the Figure caption

l. 254-256 : If last part of tge sentence refers to the distribution width , to which characteristic of the distribution is the beginning about ?

l. 273 : Don't start a sentence by "And"

l. 279 : "by the higher number of missed data points"

l. 283 : Maybe introduce the lensing effect earlier when the authors first mention it.

l. 295-296. I am not sure to really understand this sentence.

l. 304 "$E_{ab}$ values"

l. 306 "about 6.14 times larger than " ?

l. 321 : Don't start a sentence by "And"

l. 322 : It seems to me that Compact and fractal don't match together

l.357 : "overestimated or underestimated"

l.366 : "B C" remove space

l. 371 : Maybe introduce which coating material have been tested before using "these three components"

l. 373 : Is there a reason to use "As" in this sentence ?

l. 377 : "thickly coated by BrC are more numerous" maybe ?

l. 381 : Please introduce RE in the beginning of the sentence

l. 377 : Need to precise that $E_{ab}$ was studied as a function of the diameter Dc to understand the "at first", and the numbers that were cited in this sentence.

l.396 : "The oversimplification […] is inappropriate" This part of the sentence is not very clear, could you reformulate please ?

---

## Author Response (AR2)

**RESPONSE LETTER OF REVISION (EGUSPHERE-2024-3781)**

Title: Numerical quantitation on the effect of coating materials on the mixing state retrieval accuracy of fractal black carbon based on single particle soot photometer

Dear Editor:

We have revised our manuscript based on the comments of Referees. The corrections and modifications have been included in the revised manuscript and the details are listed as follows. The responses are highlighted in blue font. The changes made in the revised manuscript are marked in red font.

**Referee #4**

The author has adequately addressed the reviewers' suggestions, and the manuscript is now close to a generally publishable state. Below are my comments for further refinement before publication:

Response:

Thank you very much for reviewing out manuscript and the comments. We have responded to all the comments point by point, and the related descriptions have been modified in the revised manuscript.

(1) Expressions and discussions requiring correction: SP2 is not the only instrument capable of measuring BC mixing states. Other methods include single particle aerosol mass spectrometry (SPAMS) and single-particle soot mass spectrometry (SP-AMS).

Response:

Thanks a lot for the suggestions. We have modified the related descriptions in the revised manuscript as follows:

"The "mixing state" is a key microphysical property for aged BC, describing the mixing structure of BC and its coating. It can be characterized through different principles and instruments, such as the single particle soot photometer (SP2), single

particle aerosol mass spectrometry (SPAMS), and single-particle soot mass spectrometry (SP-AMS) (Liu et al., 2023b; Liu et al., 2025). The SP2 (Droplet Measurement Technology, Inc.), as one of the most effective online instruments that measures the mixing state of coated BC based on the combination of laser-induced incandescence and light scattering measurement, has been widely employed in laboratory and field observations (Liu et al., 2020a; Liu et al., 2022; Schwarz, 2019)."

(2) Conclusion section is overly verbose, repeating many minor results. It is recommended to focus on the core findings and implications directly related to the paper's theme, providing concise conclusions.

Response:

Thank you very much for the constructive comments. We have reduced some descriptions in the Conclusion section to further focus on the main findings of our study. The modified Conclusion is shown in the revised manuscript.

(3) Numerical precision issues: The manuscript contains numerous measured and simulated values with insufficient significant figures. These should be adjusted to reflect the precision of the measurements and simulations.

Response:

Thank you for the valuable comments. To be consistent with the measurement results of the SP2, we have modified the numerical precision in the revised manuscript. All these values of morphological parameters, optical properties, radiative effects, and the preset and retrieved mixing states have been retained to two decimal places. Except for one of the values of the BC volume fraction, 0.075.

**Referee #5**

General comments:

This paper analyses the effect of the simplified hypothesis underlying the SP2 data processing used to determine the mixing state of black carbon. In particular, the authors tested the impact of using different coating materials with different refractive indices (sulphates, organic carbon and brown carbon) and of giving the coated BC particles different shapes (ranging from more fractal-like to a core-shell model) on the mixing state. The mixing state is defined as the diameter ratio between the BC core and the whole BC particle, and is calculated from SP2 measurements using the usual retrieval method. This method uses Mie theory and assumes a core-shell morphology and a fixed refractive index. The authors identified biases resulting from these simplifications, particularly for thinly coated BC. BC coated with BrC also exhibited higher inaccuracies due to its absorbing properties. The authors further investigated subsequent errors in radiative forcing and absorption enhancement. Since the SP2 is the only online instrument that can measure the coating thickness of BC-containing particles at a particle-by-particle level, the methodology to process its mixing state data is crucial for interpreting the results. This study provides valuable insights into errors in the BC mixing state resulting from core-shell and fixed refractive index assumptions. Furthermore, the results suggest that a more accurate characterization of BC optical and radiative properties would be achieved by considering more complex morphologies and variable refractive indices. The tests and experiments conducted in this study are robust and well described. However, the good scientific quality of the paper is still suffering from a significant number of grammatical errors. Some of them are listed bellow. I would strongly recommend that the authors carefully review the entire manuscript in terms of English writing, before publication.

Response:

Thanks a lot for reviewing our manuscript and your recognition of our study. We have tried our best to carefully review the entire manuscript to correct the grammatical errors and improve the English writing. All these related modifications are given in the revised manuscript.

Specific comments:

l 14. I would add particles: coated black carbon particles (BC)

l 14. "optical properties"

l. 16. "the coating material"

l. 16 "diverse"

l. 21. "are used to study optical properties…"

l. 27 "be up to 6.15 times higher than"

l 33-34 This sentence is not clear since human = anthropogenic. Please reformulate.

L 36 "optical absorption properties"

l. 39 "BC particles are usually"

l. 40 "BC particles can be coated by various"

l. 42: "through condensation and coagulation"

l. 47 "one of the most effective online instrument that can measure"

l. 57 It would be useful to precise that the change in the Gaussian pattern is due to the different refractive indices between the coating and the core.

l. 58 : Please provide references for the refractive indices mentioned here.

l.59 : I would start a new sentence from the word "since" to add clarity.

l.60 "by assuming spherical particles and using BC density …"

l.64 "into bare-to-thinly coated"

l. 69 The work package is a bit weird here, please reformulate

l 74. "sulfate or organics"

l. 79 "the measurements of the SP2 are usually…"

l. 83 "are built to represent"

l. 84 "can be classified using the lag time"

l. 100 "is the gyration radius"

l. 100 formatting of the ith

l. 109 Please don't start a sentence with "and"

l. 112 "are built". That would be nice to add a schematic of the typical shapes of these two different models.

l. 135 "For the quantification"

Formula 6: Please define what is $X^2$. More generally, it would be better to introduce each before its corresponding definition (equations 9, 10, 11,12,13), as the authors did for i.e. equation 7.

L 150-151 This sentence is not grammatically correct, or something is missing maybe?

L 152 "the precision in the estimated typical optical"

l. 163-164 Keep the same structure, either first variable and then definition as the beginning of the sentence or the other way around

l. 176 "the variation of complex refractive"

l. 178-180 The sentence starting with "while" is not grammatically correct.

l. 184-187 The sentence starting with "as can be" is not grammatically correct.

l.189 "more compact and the retrieval"

l. 189 "BC volume fraction due to a smaller amount"

l. 192 "SP2 has a better performance"

l. 193 This sentence is not clear to me. Please reformulate.

l.198 "from about 200 to 400 nm, as shown by Liu et al."

l.201 "missing data points of retrieved"

l.202-203 Not only that but it will also lead to a bias toward size ranges of coated BC that the SP2 is able to characterize when considering the ensemble of BC particle.

Figure 1: This is good to include a schematic to show the corresponding morphologies of BC but I don't really understand what values of Dc/Dp and Dc,v each of the representation refers to. Maybe move the images more to the corners, ie. Maximum value of Dc/Dp?

Figure 3 : The Dp,v/Dc,v in vertical on the right hand side is a bit difficult to understand. Maybe move to the left, add a space between it and the y-label, and increase the font? Same for the column titles, increase font and space between title and figures

Figure 4: Describe what is the dashed line on the violins

Caption of Figure 5: "…and the corresponding core-shell model, , represented for different values of Dp,v/Dc,v."

Caption of Figure 6: Same as previous comment

l. 207 "to the coating components, thus the"

l. 211 move the "compared with the BC refractive index" to the end of the sentence

l. 212 "on the other hand, when the soot"

l. 215 Do you mean Cappa study, by "this study"? Then citation needs to be done properly: Cappa et al found ...

l. 215: The sentence starting with "The enlargement" is not clear to me. Please reformulate.

l. 220 This was already said l193, so maybe change to "that confirms..."

l. 230 "between sulfate and OC could explain their similar Dp/Dc"

l. 231: If "could" is used for the first part of the sentence, then one would use "may have caused" or so

l. 240 A higher accuracy than for other coating materials?

l. 242 "The coupling effect […] would lose": not grammatically correct. Would lead to …

l. 247 "Three distinct regions coloured in"

l. 248-250 : It would be nice to put that in the Figure caption

l. 254-256: If last part of the sentence refers to the distribution width, to which characteristic of the distribution is the beginning about?

l. 273: Don't start a sentence by "And"

l. 279: "by the higher number of missed data points"

l. 283: Maybe introduce the lensing effect earlier when the authors first mention it.

l. 295-296. I am not sure to really understand this sentence.

l. 304: "Eab values"

l. 306 "about 6.14 times larger than"?

l. 321: Don't start a sentence by "And"

l. 322: It seems to me that Compact and fractal don't match together

l.357: "overestimated or underestimated"

l.366: "B C" remove space

l. 371: Maybe introduce which coating material have been tested before using "these three components"

l. 373: Is there a reason to use "As" in this sentence?

l. 377: "thickly coated by BrC are more numerous" maybe?

l. 381: Please introduce RE in the beginning of the sentence

l. 387: Need to precise that Eab was studied as a function of the diameter Dc to understand the "at first", and the numbers that were cited in this sentence.

l. 396: "The oversimplification […] is inappropriate" This part of the sentence is not very clear, could you reformulate please?

Response:

Thank you very much for the responsible review on our manuscripts. The related descriptions about all these above 75 valuable suggestions have been rewritten or modified in the revised manuscript, and the Figures have also been made necessary modifications to improve clarity.